# New Last Glacial Maximum Ice Thickness constraints for the Weddell Sea Embayment, Antarctica

Keir A. Nichols[1], Brent M. Goehring[1], Greg Balco[2], Joanne S. Johnson[3], Andrew S. Hein[4], Claire Todd[5]

[1]. Department of Earth and Environmental Sciences, Tulane University, New Orleans, 70118, LA, USA.
[2]. Berkeley Geochronology Center, 2455 Ridge Road, Berkeley, 94709, CA, USA.
[3]. British Antarctic Survey, Natural Environment Research Council, High Cross, Madingley Road, Cambridge, CB3 0ET, UK.
[4]. School of GeoSciences, University of Edinburgh, Drummund Street, Edinburgh, EH8 9XP, UK.
[5]. Department of Geosciences, Pacific Lutheran University, Tacoma, 98447, WA, USA.

*Correspondence to:* Keir Nichols (knichol3@tulane.edu)

**Abstract.** We describe new Last Glacial Maximum (LGM) ice thickness constraints for three locations spanning the Weddell Sea Embayment (WSE) of Antarctica. Samples collected from the Shackleton Range, Pensacola Mountains, and the Lassiter Coast constrain the LGM thickness of the Slessor Glacier, Foundation Ice Stream, and grounded ice proximal to the modern Ronne Ice Shelf edge on the Antarctic Peninsula, respectively. Previous attempts to reconstruct LGM-to-present ice thickness changes around the WSE used measurements of long-lived cosmogenic nuclides, primarily $^{10}$Be. An absence of post-LGM apparent exposure ages at many sites led to LGM thickness reconstructions that were spatially highly variable, and inconsistent with flowline modelling. Estimates for the contribution of the ice sheet occupying the WSE at the LGM to global sea level since deglaciation vary by an order of magnitude, from 1.4 to 14.1 m of sea level equivalent. Here we use a short-lived cosmogenic nuclide, in situ produced $^{14}$C, which is less susceptible to inheritance problems than $^{10}$Be and other long-lived nuclides. We use in situ $^{14}$C to evaluate the possibility that sites with no post-LGM exposure ages are biased by cosmogenic nuclide inheritance due to surface preservation by cold-based ice and nondeposition of LGM-aged drift. Our measurements show that the Slessor Glacier was between 310 and up to 655 m thicker than present at the LGM. The Foundation Ice Stream was at least 800 m thicker, and ice on the Lassiter Coast was at least 385 m thicker than present at the LGM. With evidence for LGM thickening at all of our study sites, our in situ $^{14}$C measurements indicate that the long-lived nuclide measurements of previous studies were influenced by cosmogenic nuclide inheritance. Our inferred LGM configuration, which is primarily based on minimum ice thickness constraints and thus does not constrain an upper limit, indicates a relatively modest contribution to sea level rise since the LGM of <4.6 m, and possibly as little as <1.5 m.

## 1. Introduction

We describe new constraints on Last Glacial Maximum (LGM, ca. 26 to 15 ka; Peltier and Fairbanks, 2006) ice thickness changes from three locations within the Weddell Sea Embayment (WSE) of Antarctica (Fig. 1). The WSE drains approximately one fifth of the total area of the Antarctic ice sheets (AIS) (Joughin et al., 2006) and is thus an important contributor to LGM-to-present and, potentially, future sea level change. Previous attempts to reconstruct LGM-to-present ice thickness changes around the WSE used measurements of long-lived cosmogenic nuclides, primarily $^{10}$Be (half-life 1.387 ± 0.012 Ma; Chmeleff et al., 2010; Korschinek et al., 2010) and $^{26}$Al (half-life 705 ± 17 ka; Norris et al., 1983), sourced from bedrock and erratic cobbles proximal to modern glacier surfaces. Through measuring the cosmogenic nuclide concentration of samples of glacial deposits and bedrock, one can constrain the magnitude and timing of past changes in the thickness of adjacent ice masses. However, an absence of post-LGM apparent exposure ages at many sites around the WSE led to LGM thickness reconstructions that were spatially highly variable, and inconsistent with flowline modelling (e.g. Whitehouse et al., 2017). Consequently, estimates based on ice models constrained by field evidence (Le Brocq et al., 2011) and by relative sea level records and earth viscosity models (Bassett et al., 2007) for the contribution of the sector to global sea level since

deglaciation began vary by an order of magnitude, from 1.4 to 14.1 m, respectively. The lack of geological evidence for LGM thickening is also manifest in a misfit between present day geodetic uplift rate measurements in southern Palmer Land and predicted uplift rates from a glacial isostatic adjustment (GIA) model (Wolstencroft et al., 2015). Constraining the previous vertical extent of ice provides inputs to numerical models investigating both the response of the ice sheet to past
and potential future changes in climate and sea level (e.g. Briggs et al., 2014; Pollard et al., 2016, 2017; Whitehouse et al., 2017), as well as the response of the solid earth to past ice load changes to quantify present day ice-mass loss (e.g. Wolstencroft et al., 2015). Furthermore, quantifying the LGM dimensions of the WSE sector of the AIS is required to further constrain the offset between estimates for post-LGM sea level rise and estimates of the total amount of ice melted since the LGM. The former is sourced from sea level index points, and the latter is sourced from our knowledge of the dimensions of
ice masses at the LGM (Simms et al., 2019). Currently, the "missing ice" accounts for between $15.6 \pm 9.6$ m and $18.1 \pm 9.6$ m of global sea level rise since the LGM (Simms et al., 2019).

   Although the use of cosmogenic nuclide geochronology to study the AIS is clearly proven (e.g. Stone et al., 2003; Ackert et al., 2007), applications in the WSE are challenging. Many studies, despite making multiple cosmogenic nuclide measurements from relatively large numbers of samples, observed no or few post-LGM exposure ages (Hein et al., 2011,
2014; Balco et al., 2016; Bentley et al., 2017). With no evidence for LGM ice cover, it was not clear whether sites were covered by ice at the LGM, or whether sites were covered but the ice left no fresh deposits on top of those yielding pre-LGM ages. It is therefore currently unknown whether ice was thicker than present during the LGM at the Schmidt Hills in the Pensacola Mountains, and in the Shackleton Range (Figs. 1 and 2). Results from the Schmidt Hills (Fig. 2) indicating no LGM thickening of the Foundation Ice Stream (FIS) are particularly problematic, as thickening of 500 m from the Williams
Hills, 50 km upstream of the Schmidt Hills, produces a LGM surface slope that is steeper than glaciological models permit and is also steeper than present-day ice surface slopes (Balco et al., 2016). Cold-based ice and an associated lack of subglacial erosion is the likely cause of the complex [10]Be data sets, evidenced by numerous studies in the WSE that report [10]Be and [26]Al ratios significantly below those predicted for continuous exposure which is indicative of significant periods of non-erosive burial (e.g. Bentley et al., 2006; Sugden et al., 2017). Cold-based ice preserves surfaces (e.g. Stroeven et al.,
2002; Sugden et al., 2005; Gjermundsen et al., 2015), allowing nuclide concentrations to persist within surfaces from previous periods of exposure to the present, a phenomenon known as inheritance. Long-lived nuclides are particularly susceptible to inheritance due to their long half-lives which, when protected from erosion beneath cold-based ice, require long periods of burial to reduce concentrations to below measurable levels. When covered by cold-based ice during glaciations, concentrations of long-lived nuclides record exposure during multiple separate ice free periods rather than just
the most recent one. Inheritance thus hinders interpretations of cosmogenic nuclide measurements.

   We resolve conflicting LGM thickening estimates based on [10]Be measurements by using measurements of in situ produced [14]C, a cosmogenic nuclide that is, owing to a short half-life of 5730 years, largely insensitive to inheritance. We present the in situ [14]C analysis of transects of erratic and bedrock samples from the Shackleton Range, Lassiter Coast and Pensacola Mountains (Fig. 1). Our results constrain the LGM thickness of the Slessor Glacier to between 310 and up to 655
m. We show that ice was at least 385 m thicker than present during the LGM at the Lassiter Coast, proximal to the modern Ronne Ice Shelf edge. Our data also constrain the LGM thickness of the FIS to at least 800 m at the Schmidt Hills. Replicate measurements made from four samples revealed higher than expected variability of in situ [14]C measurements, which is discussed in Sect. 4.1. Our thickness estimates are comparable to those of Hein et al. (2016) in the Ellsworth Mountains, as well as those of Balco et al. (2016) and Bentley et al. (2017) in the Williams and Thomas Hills. Although our results show
that locations around the WSE were buried by hundreds of metres of ice, this is less than called for by some reconstructions. Our inferred LGM configuration, which is primarily based on minimum ice thickness constraints and thus does not constrain an upper limit, indicates a relatively modest contribution to sea level rise since the LGM of <4.6 m, and possibly as little as <1.5 m.

**1.1 The Last Glacial Maximum in the Weddell Sea Embayment**

   Although it is clear that grounded ice in the WSE has been thicker in the past (Bentley and Anderson, 1998), there is little evidence as to the thickness and grounding line position of the ice sheet at the LGM, with contrasting evidence from marine sources, and those inferred from terrestrial studies (Hillenbrand et al., 2014). Terrestrial evidence for the extent of ice

in the WSE during the LGM takes the form of numerous cosmogenic nuclide studies. Bentley et al. (2006) measured the [10]Be and [26]Al content of erratics on the southern Antarctic Peninsula. Studies report cosmogenic nuclide concentrations from the Meyer Hills, Patriot Hills, Marble Hills, and the Flower Hills, all in the Ellsworth Mountains (Bentley et al., 2010; Fogwill et al, 2014; Hein et al., 2016; Sugden et al; 2017),  the Pensacola Mountains (Hodgson et al, 2012; Balco et al, 2016; Bentley et al, 2017), and the Shackleton Range (Fogwill et al., 2004; Hein et al, 2011, 2014). Figure 2 summarises the ice thickness estimates from these studies. The majority of estimates are sourced from [10]Be measurements, with some accompanying [26]Al measurements. Two exceptions are Fogwill et al. (2014) and Balco et al. (2016), whom combined some in situ [14]C measurements with [10]Be measurements to constrain the thickness of the Rutford and Institute ice streams and the Foundation Ice Stream, respectively. The highest elevation post-LGM exposure ages at each site delineate the minimum vertical extent of ice at the LGM. Ice thickness estimates vary spatially around the embayment, ranging from zero to hundreds of metres of LGM thickening.

Marine geological and geophysical evidence in the southern Weddell Sea indicates a significantly expanded WSE LGM configuration, with subglacial till, subglacial bedforms and a grounding zone wedge found towards the shelf edge (Hillenbrand et al., 2012, 2014; Larter et al., 2012; Arndt et al., 2017). As a result, there is currently a disconnect between marine evidence for a greatly expanded WSE sector and terrestrial evidence indicating little to no vertical change at the LGM in some areas. Hillenbrand et al. (2014) propose two potential LGM configurations of the WSE sector of the AIS. The first scenario, based on terrestrial evidence for vertical LGM ice thicknesses, involves a complex configuration with the grounding line of the ice sheet situated towards the shelf edge and a largely ice-free Filchner Trough and western margin of the WSE. The second scenario, based on marine evidence, places the grounding line of the ice sheet at the shelf edge across the width of the WSE. Flowline modelling of the response of the FIS, which occupied the Filchner Trough at the LGM, to the onset of glacial conditions shows that there are two plausible LGM grounding line positions for the ice stream: one situated at the shelf edge, and another at the northern margin of Berkner Island (Whitehouse et al., 2017).

## 1.2 In situ [14]C exposure dating

Cosmogenic nuclides [10]Be and [26]Al have half-lives that are much longer than glacial-interglacial cycles, so [10]Be and [26]Al concentrations produced in previous interglacials persist to the present if buried by non-erosive, cold-based ice. The short half-life of in situ [14]C means that only short periods of burial are required to significantly reduce concentrations from previous periods of exposure, making in situ [14]C less sensitive to inheritance than longer-lived nuclides. For example, a burial duration beneath non-erosive, cold-based ice of 11 kyr results in ca. 74% of the original in situ [14]C decaying away. Furthermore, continuously exposed, slowly eroding surfaces reach an equilibrium between production and decay of in situ [14]C ("saturation") after approximately 30 to 35 kyr. A sample that has reached saturation thus requires low erosion and continuous exposure from before the LGM, whilst a sample that yields a concentration below saturation requires ice cover during the last ca. 35 kyr. Surfaces yielding saturation concentrations therefore provide an upper limit on LGM thickening. Figure 3 shows a hypothetical ice surface elevation change history at a nunatak partially buried by cold-based, non-erosive ice during the LGM, with associated in situ [14]C measurements from samples collected along an elevation transect on the surface of the nunatak. There is a transition from undersaturated to saturated samples, a discontinuity in the [14]C concentrations which constrains the LGM ice thickness. The "true exposure" data points represent in situ [14]C concentrations with resulting exposure ages matching the post-LGM ice-surface lowering history. The "apparent exposure" data points were saturated at the onset of ice cover and include in situ [14]C that persists to the present due to an insufficient amount of time passing for it to decay away. For the five undersaturated samples, which were buried by ice for differing durations, a range of ~2 to ~4 % of the [14]C accumulated prior to burial will persist to the present. In terms of the effect on resulting exposure ages, the sample exposed at 10 ka yields an apparent exposure age of 11.41 ka (~13 % increase), and the sample exposed at 2 ka yields an apparent exposure age of 2.17 ka (~8 % increase). Without knowing the burial duration of the samples or whether or not the samples were indeed saturated upon burial by LGM ice, we do not know the exact quantity of in situ [14]C inherited in the samples. The in situ [14]C exposure ages are therefore maximum deglaciation ages. In the same hypothetical scenario with the same samples, ca. 98 % and 97 % of the [10]Be and [26]Al accumulated prior to burial will persist to the present, respectively.

We report in situ $^{14}$C concentrations measured from both erratic and bedrock samples, with primarily erratic samples from the Shackleton Range and the Pensacola Mountains, and solely bedrock from the Lassiter Coast. We assume both materials provide the same information regarding the timing of ice retreat and constraining LGM ice thicknesses. For example, we assume that both erratics and bedrock samples saturated with in situ $^{14}$C indicate that their respective sampling locations were ice free for the last 30 to 35 kyr. With the exception of two samples, all of our erratic samples have previously been measured for their $^{10}$Be content (Hein et al., 2011, 2014, Balco et al., 2016), with the vast majority yielding ages far in excess of the LGM. It is highly likely that these erratic samples have been repeatedly covered and exposed by cold-based ice. Having been covered and uncovered in situ, the erratic samples can thus effectively be considered bedrock. There are, however, potential situations where our assumption that bedrock and erratic samples provide the same information with respect to the timing of changes in ice thickness is not met and resulting $^{14}$C concentrations misrepresent the age of deglaciation, creating scatter in the measured in situ $^{14}$C data. Erratic samples may, for example, be sourced from mass movement onto glacier surfaces, producing spuriously high $^{14}$C concentrations (See Balco et al., 2019). Spuriously high, in excess of saturation, in situ $^{14}$C concentrations sourced from bedrock samples, however, can only result from analytical errors and thus provides an important test for the premise of the technique. Additionally, erratic cobbles may have undergone downslope movement post-deposition and may have flipped over, or may have been subjected to high erosion rates, which could produce in situ $^{14}$C concentrations with resulting exposure ages lower than the true age of deglaciation. Snow shielding of sample locations is another mechanism leading to exposure ages which underestimate the age of deglaciation and can influence both bedrock and erratic samples. Whilst not without challenges, our in situ $^{14}$C measurements provide an opportunity to unambiguously show whether sites around the WSE were covered by ice at the LGM.

## 1.3 Sample Sites

### 1.3.1 Shackleton Range

The Shackleton Range is located in Coats Land in northeastern WSE, adjacent to Slessor Glacier (Figs. 1 and 4a). Slessor Glacier drains ice from the East Antarctic Ice Sheet (EAIS) into the Filchner Ice Shelf. Mt. Skidmore is located approximately 25 km upstream of the modern Slessor Glacier grounding line, with the Köppen and Stratton glaciers respectively joining the Slessor Glacier to the north and south of Mt. Skidmore (Fig. 4a). Proximal to sampling locations are Ice Tongue A and Ice Tongue B of the Stratton Glacier, and the Snow Drift Glacier (Fig. S1). We assume that samples collected from Mt. Skidmore record changes in the thickness of the Slessor Glacier. However, it is possible that samples collected proximal to the smaller ice masses may have been buried by them, rather than by the Slessor Glacier, potentially complicating the interpretation of results. The modern Slessor Glacier surface is situated at ~200 m a.s.l. proximal to Mt. Skidmore, with exposed surfaces of Mt. Skidmore located up to over ~820 m a.s.l. Mt. Provender is located adjacent to the Slessor Glacier grounding line and is bounded by the Stratton and Blaiklock glaciers to the north and south, respectively. Exposed rock of Mt. Provender rises from the modern ice surface up to over ~900 m a.s.l. We analysed 11 samples from the Shackleton Range (Table S1), with two from Mt. Provender and nine from Mt. Skidmore (Fig. 4a). At Mt. Provender we analysed one erratic cobble from near the modern ice surface and one bedrock sample from ~650 m above it (Fig. S2). Samples from Mt. Skidmore include one bedrock sample and eight cobbles that form an elevation transect from near the modern ice surface to ~300 m above it (Fig. S1). The two highest elevation samples collected from Mt. Skidmore are proximal to the main trunk of the Stratton Glacier more so than the Slessor Glacier, and were collected from ca. 115 and 130 m above the modern Stratton Glacier surface. The two highest elevation samples on Mt. Skidmore therefore may represent a Stratton Glacier ice surface lowering more so than the Slessor Glacier, and thus are presented as a separate sample group to those collected proximal to the Slessor Glacier.

### 1.3.2 Lassiter Coast

The Lassiter Coast is located on the east coast of southern Palmer Land, adjacent to the present position of the Ronne Ice Shelf edge (Fig. 1). The modern ice surface is situated at 490 m a.s.l. Johnson et al. (2019) collected samples from several sites in this area (Fig. 4b) and carried out $^{10}$Be measurements; we subsequently carried out $^{14}$C measurements on these samples as part of the present study, and the $^{14}$C results are reported both here and in Johnson et al. (2019). Here we discuss results for a total of eight bedrock samples from Mt. Lampert and the Bowman Peninsula collected from 20 to 385 m

above the modern ice surface (Figs. 4b and S3); see Table S1 for sample data and Johnson et al. (2019) for [10]Be measurements. The adjacent Johnston Glacier drains ice from central Palmer Land into the WSE (Fig. 4b). We interpret the samples together as effectively a single elevation transect that records changes in the thickness of grounded ice in the WSE immediately east of these sites after the LGM.

### 1.3.3 Pensacola Mountains

The Schmidt Hills are a series of nunataks adjacent to the FIS in the southeast WSE (Figs. 1 and 4c) proximal to the modern grounding line. The FIS is a major ice stream that drains ice from both the EAIS and West Antarctic Ice Sheet (WAIS) into the WSE. The surface of the FIS adjacent to the Schmidt Hills is situated ca. 200 m a.s.l., with exposed surfaces of the Schmidt Hills reaching up to 1100 m a.s.l. The Thomas Hills are another series of nunataks adjacent to the FIS,
located ~130 km upstream of the Schmidt Hills (Figs. 1, 4d). The main trunk of the FIS adjacent to the Thomas Hills is near 550 m a.s.l., with the Thomas Hills rising up to 1050 m a.s.l. The local ice margin of the FIS at the Thomas Hills is situated ~75 m below the centre of the FIS. We analysed 17 samples from the Pensacola Mountains (Table S1); 15 from the Schmidt Hills and two from the Thomas Hills. We made a further seven repeat measurements from four samples collected from the Schmidt Hills. Samples from the Schmidt Hills were collected from Mount Coulter and No Name Spur (Figs. 4c and S4)
from close to the modern ice surface to approximately 800 m above it. We also analysed two samples from the Thomas Hills which were collected from Mount Warnke ca. 320 m above the FIS ice margin (Figs. 4d and S5). The highest elevation sample from the Schmidt Hills, collected from ca. 1035 m a.s.l., is the only bedrock sample analysed from the Pensacola Mountains, with the rest being erratic cobbles.

## 2. Methods

We used between 0.5 and 10 g of quartz from each sample for in situ [14]C analysis. The methodology used for the isolation of quartz varies for samples from different sample sites because quartz was previously isolated for prior cosmogenic nuclide studies (see Hein et al., 2011; Balco et al., 2016). For samples processed at the Tulane University Cosmogenic Nuclide Laboratory (primarily those from the Lassiter Coast), quartz was isolated through crushing, sieving, magnetic separation and froth flotation (modified from Herber, 1969) of sample material. Samples were then etched for at
least two periods of 24 hours on both a shaker table in 5 % $HF/HNO_3$ and then in an ultrasonic bath in 1 % $HF/HNO_3$. This leaching procedure removes the organic compound laurylamine used in the froth flotation procedure (Nichols and Goehring, in review) that could otherwise potentially contaminate our samples with modern carbon.

Carbon was extracted using the Tulane University Carbon Extraction and Graphitization System (TU-CEGS), following the method of Goehring et al. (2019). Quartz is step-heated in a lithium metaborate ($LiBO_2$) flux and a high-purity
$O_2$ atmosphere, first at 500 °C for 30 minutes, then at 1100 °C for three hours. Released carbon species are oxidised to form $CO_2$ via secondary hot-quartz-bed oxidation, followed by cryogenic collection and purification. Sample yields are measured manometrically, and samples are diluted with [14]C-free $CO_2$. A small aliquot of $CO_2$ is collected for $\delta^{13}C$ analysis, and the remaining $CO_2$ is graphitised using $H_2$ reduction over an Fe catalyst. We measured [14]C/[13]C isotope ratios at either Lawrence Livermore National Laboratory Center for Accelerator Mass Spectrometry (LLNL-CAMS) or Woods Hole National Ocean
Sciences Accelerator Mass Spectrometry (NOSAMS) (Table S2). Stable carbon isotope ratios were measured at the UC-Davis Stable Isotope Facility.

Apparent exposure ages were calculated using v. 3 of the online calculators formerly known as the CRONUS-Earth online calculators (Balco et al., 2008). The online calculators use the production rate scaling method for neutrons, protons and muons of Lifton et al. (2014) (also known as LSDn). We use repeat measurements of the in situ [14]C concentration of the
CRONUS-A interlaboratory comparison standard (Jull et al., 2015; Goehring et al., 2019) to calibrate the [14]C production rate. We assume CRONUS-A is saturated with respect to in situ [14]C, given that, based on geological mapping and an ash chronology, the sampling location has remained ice-free since >11.3 Ma (Marchant et al., 1993). All reported in situ [14]C measurements from CRONUS-A, made at multiple laboratories, yield concentrations equivalent to saturation based on other calibration data from elsewhere in the world (e.g. Jull et al., 2015; Fülöp et al., 2019; Goehring et al., 2019; Lamp et al.,
2019). We use the CRONUS-A measurements to calibrate the [14]C production rate to reduce scaling extrapolations. Repeat

measurements of both CRONUS-A and other samples using the TU-CEGS show that the reproducibility of in situ [14]C measurements is approximately 6 %. We therefore use a 6 % uncertainty for our measured in situ [14]C concentrations when calculating exposure ages, as this exceeds the reported analytical uncertainty for all of our in situ [14]C measurements. Ages are included in Table S2 for completeness but are primarily discussed in the text as either finite or infinite ages. Infinite ages are those for which the measured concentration is above the uncertainty of the saturation concentration for the elevation of a given sample.

We made seven replicate measurements from four samples from the Schmidt Hills that initially yielded saturation or near-saturation in situ [14]C concentrations. We made the first four replicate measurements using the same samples to test the validity of the saturation or near-saturation initial measurements. The second set of measurements produced in situ [14]C concentrations below saturation. Given the difference between the initial measurements and the replicates, we made a further three measurements from three of the same four samples.

## 3. Results

The vast majority of the [10]Be ages reported by Hein et al. (2011, 2014) in the Shackleton Range exceed 100 ka, whilst we find finite [14]C ages at both Mt. Skidmore and Mt. Provender (Figs. 5 and S6). At Mt. Skidmore, finite ages are evident across the entire Mt. Skidmore transect, including those sampled proximal to the Stratton Glacier (Fig. 5). Samples were collected from multiple ridges of Mt. Skidmore and thus would not necessarily be expected to form a single age-elevation line. The uppermost sample proximal to the Slessor Glacier, collected ~310 m above the modern ice surface, provides a lower limit for the LGM ice thickness of the ice mass. The two samples proximal to the Stratton Glacier, an erratic and bedrock sample with ~17 m a.s.l. between them, are indistinguishable from one another within uncertainties and constrain the LGM thickness to at least 130 m thicker than present. At Mt. Provender, one sample collected proximal to the local Slessor Glacier margin yields a finite age. A second sample from ~890 m a.s.l. (~655 m above the modern ice surface) yields an infinite age, placing an upper limit on the LGM thickness at ~655 m larger than present. We note that the upper limit of 655 m is based on a single in situ [14]C measurement and discuss this limitation further in Sect. 4.2. If quartz was available for additional samples previously collected from Mt. Provender (Hein et al, 2011, 2014), then further measurements could have been made to validate this measurement. The quartz was, however, exhausted in the process of measuring long-lived nuclides. One sample from Mt. Skidmore, collected from ~284 m a.s.l., yields an infinite age. Above the saturated sample we observe seven finite-aged samples which require significant periods of burial beneath ice to account for their in situ [14]C concentrations. It is glaciologically impossible to have the sample at ~284 m a.s.l. exposed for ca. 35 kyr whilst those above it were covered presumably by the Slessor and Stratton glaciers. The infinite age of the sample could be due to scatter within the [14]C measurements, and the fact that the sample is an erratic does allow the possibility of an unlikely geomorphic scenario. As described in Sect. 1.2, erratic samples may be sourced from mass movement onto glacier surfaces, producing spuriously high [14]C concentrations (Balco et al., 2019).

On the Lassiter Coast, Johnson et al. (2019) report [10]Be ages which, with the exception of three measurements, all exceed ~100 ka, whilst all of the in situ [14]C ages are finite and fall within the Holocene (Figs. 6 and S7). The associated in situ [14]C concentrations are similar over the range of sample elevations (Fig. 6). The uppermost sample, collected ~385 m above the modern ice surface, provides a lower limit on the thickness of LGM ice at the Lassiter Coast. The small range of ages across ca. 300 m elevation transect indicate that ice thinning occurred rapidly at this study site (Johnson et al., 2019).

At the Schmidt Hills, [10]Be ages from Balco et al. (2016) and Bentley et al. (2017) range from ~140 ka to 3 Ma (Fig. 7). We observe finite ages at low elevations and finite, close to infinite, and infinite ages at higher elevations (Figs. 7 and S8). Given that higher elevations cannot be covered by ice unless lower elevations were also covered, we remeasured the apparently infinite and near-infinite aged samples (~500 to ~920 m a.s.l., or ~270 to ~690 m above the modern ice surface). The replicate results (Fig. 7) show high variability, greater than that observed in previous repeat measurements of CRONUS-A and other samples (Goehring et al., 2019). There is no apparent analytical reason for the initial measurements yielding infinite or near-infinite ages and then yielding differing concentrations with repeat measurements. Samples yielding only finite ages (those that were not measured multiple times) are observed up to ~420 m a.s.l., or ~190 m above the modern ice surface. In addition, the bedrock sample collected from ca. 1035 m a.s.l. yields a finite age, indicative of a LGM thickness at least ~800 m larger than present for the FIS at the Schmidt Hills. The agreement between the bedrock age and the finite

measurements from lower elevations means we conclude that, at the Schmidt Hills, the FIS was 800 m thicker than present at the LGM. This conclusion, and the repeat measurements with a high degree of scatter, are discussed in Sect. 4.1 and 4.2.

The two samples collected from the Thomas Hills yield finite ages within ~0.2 ka of one another (Figs. 8 and S9). Results thus indicate that the FIS was at least ~320 m thicker than present at the LGM at the Thomas Hills. The apparent in situ [14]C ages, at ~10 ka, are consistent with a cluster of [10]Be ages between 7 and 9 ka in the Thomas Hills reported by Balco et al. (2016) from 225 m above the modern FIS surface, as well as a [10]Be age of 4.2 ka reported by Bentley et al. (2017) collected 125 m above the modern ice surface. Considering the evidence for significant LGM thickening of the FIS from our in situ [14]C results from the Thomas Hills, as well as [10]Be ages of Balco et al. (2016) and Bentley et al. (2017) from both the Williams and Thomas Hills, we infer that it is likely that the FIS reached up to 800 m above its present thickness at the LGM at the Schmidt Hills. We discuss this inference further in the following section.

## 4. Discussion

### 4.1. Assessment of [14]C elevation transects

The premise of our study is that one can clearly infer if a site was ice-covered at the LGM by determining whether the in situ [14]C concentration of samples from that site are at or below saturation. In this section we assess the success of the approach. To assess the validity of this method, we can, for example, identify where the in situ [14]C data records ice thinning, with saturated samples or the oldest exposure ages at the highest elevations and a trend of decreasing in situ [14]C age toward modern ice surfaces. Consistency between in situ [14]C data and other nuclide concentrations (e.g. [10]Be) could also help validate the in situ [14]C measurements. We also look at factors beyond the in situ [14]C concentrations, such as the glaciological link between study sites, which may add clarity where the in situ [14]C measurements show a high degree of scatter.

At some sites our results are consistent with the premise, as well as internally consistent. At the Lassiter Coast, ages decrease toward the present ice sheet surface. Though limited by the number of samples, two measurements from Mt. Provender align with the premise of our study, in that we find a finite age located at a low elevation with an infinite age above it. In the Thomas Hills we see consistency between the finite [14]C ages and previously published [10]Be ages (Balco et al., 2016; Bentley et al., 2017). Fogwill et al. (2014) also observe consistency between [14]C and [10]Be ages, which constrain the LGM thickness and dynamics of the Rutford Ice Stream. However, we observe apparently finite ages above apparently infinite ages at the Schmidt Hills, a scenario that is glaciologically impossible if our assumptions are correct that samples are indeed glacial erratics that have either been deposited previously and repeatedly covered by cold-based ice or delivered to their sampling location during the last glaciation and were sourced subglacially. The scatter observed in the repeat measurements (Fig. 7) is greater than that of repeat measurements made of CRONUS-A and other samples made in our laboratory (Goehring et al., 2019). Three samples from the Schmidt Hills (006-COU, 008-NNS and 046-NNS, collected from ~920, ~710 and ~500 m a.s.l., respectively) were previously measured for their in situ [14]C content and were published by Balco et al. (2016). All three of the samples previously measured by Balco et al. (2016) yielded higher concentrations (two of which were above saturation with the third at saturation) than their new measurements presented in this study. Furthermore, two of the three samples (006-COU and 046-NNS) were measured multiple times (in this study) and display the high scatter under discussion. Balco et al. (2016) proposed unrecognised measurement error as the cause of the spuriously high in situ [14]C concentrations. Why the replicate measurements from samples from the Schmidt Hills display a high degree of scatter remains to be determined.

The most likely reason for [14]C measurement error is contamination by modern [14]C, which would result in a spuriously high concentration. In contrast, a spuriously low concentration is less likely, and we are not aware of any documented instances of this. In our laboratory we have found that it is relatively easy to contaminate a sample with modern carbon through the use of organic compounds in the froth flotation mineral separation procedure (Nichols and Goehring, in review). However, froth flotation was not used to isolate the quartz of any of the samples for which replicate measurements were made. On multiple occasions we have observed spuriously high [14]C concentrations, far in excess of saturation concentrations, from quartz separates of fine grain sizes (ca. 60 μm) that were not isolated using froth flotation. We do not yet know the reason for the fine grain sizes yielding elevated [14]C concentrations, but one hypothesis is that the finite-aged replicate measurements were unintentionally made using quartz separates with a coarser average grain size than the initial

infinite measurements. We believe the above observations indicate that the increased scatter may be the result of measurement difficulties, perhaps lithology- or grain size-specific.

Regardless of the cause of the high degree of scatter observed in the replicate measurements, we need to discuss possible explanations for apparently infinite ages at lower elevations than apparently finite ages to isolate which measurements (infinite vs finite replicates) are the most valid to base interpretations on. At the Schmidt Hills, the hypothesis that infinite ages situated below finite ages are spurious and due to measurement errors is consistent with the glaciological relationship amongst the Schmidt, Thomas and Williams Hills (see Sect. 4.2) and is also consistent with the finite bedrock age sourced from a higher elevation. The bedrock age is a robust constraint because the sample cannot have been subjected to geomorphic scenarios that could cause the resulting age to misrepresent the timing of deglaciation. The hypothesis that the infinite ages are correct produces a steep LGM surface slope and is not consistent with thickness estimates from the Williams and Thomas Hills. We elaborate on this point in Sect. 4.2.

As described in Sect. 1.2, it is theoretically possible for in situ $^{14}$C saturated erratic samples to occur at lower elevations than finite ages in rare situations if the former were transported by LGM ice. Balco et al. (2019) observed an apparently saturated sample beneath finite aged samples. Supported by field observations, Balco et al. (2019) propose that the saturated sample was sourced from a rockfall upstream and transported to the study site as supraglacial debris, explaining the elevated in situ $^{14}$C concentration. Whilst this could explain the low-elevation saturated sample at Mt. Skidmore, as well as infinite measurements situated beneath finite measurements at the Schmidt Hills, it does not explain the poor reproducibility of the Schmidt Hills measurements.

We conclude that the basic concept works, as shown at the Lassiter Coast and the Shackleton Range, as well as in other aforementioned studies. In the following section we discuss the implications for LGM ice sheet reconstructions. However, it is clear that more investigation into laboratory issues and geological and geomorphic factors is required to identify the cause or causes of apparently site- or lithology-specific excess scatter in in situ $^{14}$C measurements.

**4.2 LGM ice thicknesses in the Weddell Sea Embayment**

Our LGM thickness estimates are summarised in Fig. 9. The new in situ $^{14}$C concentrations indicate that the vast majority, if not the entirety, of Mt. Skidmore, and presumably much of Mt. Provender, were covered by ice at the LGM (Figs. 5 and 10). The highest elevation samples on Mt. Skidmore proximal to the Slessor Glacier yield infinite ages and indicate that the ice stream was at least 300 m thicker at the LGM than at present. This assumes the samples were not influenced by expansion of local ice masses from the southeast (Fig. S1). If so, and assuming the surface gradient of Slessor Glacier during the LGM was similar to today, this would suggest the Slessor Glacier was ~300 m thicker at Mt. Provender at the LGM. With no high-elevation infinite ages found on Mt. Skidmore, our thickness estimates for the Slessor Glacier are likely conservative estimates. Finite ages are observed across the entire Mt. Skidmore transect and there is only a single exposed peak (between Ice Tongue A and Ice Tongue B of the Stratton Glacier, Fig. S1) that is at a higher elevation than our sampling locations (ca. 25 m higher). Presumably, given the evidence for the expansion of the Slessor and Stratton glaciers, this small peak was covered by these or local ice masses at the LGM. Our data therefore indicate that, regardless of the source, the Mt. Skidmore site was covered by ice during the LGM, whilst the top of Mt. Provender remained exposed. Whilst the upper limit of LGM ice at Mt. Provender is based on a single sample, we believe this sample is a reliable indicator of LGM ice thickness for the following reasons. The sample is sourced from bedrock and therefore cannot have been subjected to geomorphic scenarios causing the exposure age to misrepresent the timing of ice retreat. Furthermore, froth flotation, which introduces modern carbon to sample material (Nichols and Goehring, in review), was not used to isolate quartz for this sample. Our thickness constraints (~300-655 m) supersede those of previous exposure dating studies that found no evidence from long-lived isotopes for a thicker Slessor Glacier at the LGM (Hein et al, 2011, 2014). Our LGM thickness constraints for the Slessor Glacier are consistent with our other sites as well as those of previous authors for a significantly thicker FIS at the LGM (Balco et al., 2016; Bentley et al., 2017).

The new in situ $^{14}$C results from the Lassiter Coast show that bedrock surfaces 385 m above the modern ice surface were covered by ice at the LGM. As with results in the Pensacola Mountains, with only a lower limit for the LGM thickness of 360 m, there could have been thicker ice on the Lassiter Coast at the LGM. The in situ $^{14}$C measurements contrast with

[10]Be measurements that were likely influenced by cold-based ice cover, resulting in nuclide inheritance (Johnson et al., 2019). The finite in situ [14]C ages of samples collected from 628 to 875 m a.s.l., with ages between $6.0 \pm 0.7$ ka and $7.5 \pm 0.9$ ka, are consistent with a minimum age of grounded ice retreat from a marine sediment core close to the modern ice shelf edge of $5.3 \pm 0.3$ kcal yr BP (Hedges et al., 1995; Crawford et al., 1996; Fig. 1). The fact that significant thinning occurred in the Holocene may help explain the misfit between GIA models and GPS measurements in Palmer Land (Wolstencroft et al., 2015). A thicker ice load at the LGM than that used by current ice models, or present ice load estimates that persist into the Holocene, are two potential solutions postulated by Wolstencroft et al. (2015) to explain the misfit. Further work is needed to take our new ice history into account and to investigate if a minimum of 385 m of ice at the LGM and subsequent rapid thinning at ~7 ka at the Lassiter Coast can help account for the offset.

Our in situ [14]C data indicate that the FIS was at least 800 m thicker than present at the Schmidt Hills at the LGM, which contrasts with previous studies which found no evidence for the LGM thickness of the FIS at the Schmidt Hills (Balco et al., 2016; Bentley et al., 2017). We base our LGM thickness estimate on the aforementioned finite-aged repeat measurements and the finite aged bedrock sample, rather than on the poorly reproduced infinite aged-measurements. There is robust evidence for a FIS that was at least 500 m thicker than present at the LGM at the Williams Hills, located only 50 km upstream of the Schmidt Hills (Figs. 1 and 11; Balco et al., 2016; Bentley et al., 2017). Given the evidence for a significantly thicker FIS proximal to the Schmidt Hills, we argue that the repeat measurements and the bedrock measurement indicative of the FIS being 800 m thicker are glaciologically most-likely, and thus base our LGM ice thickness estimates on them. Using the infinite measurements and accompanying constraint at the Schmidt Hills for the LGM thickness of 320 m thicker than present produces a steep surface slope from the nearby Williams Hills (Fig. 11), though less so than the surface slope produced when no LGM thickening is inferred at the Schmidt Hills based on [10]Be measurements (Balco et al., 2016). The two measurements from the Thomas Hills provide a lower limit for the LGM thickness, but the possibility remains that there was more thickening than the ca. 320 m in situ [14]C constraint. Fig. 11 tentatively indicates that the FIS may have been ~900 m thicker when using the modern surface profile of the FIS increased in elevation up to the height of the finite ages from the Schmidt Hills and post-12 ka [10]Be ages from Balco et al. (2016) and Bentley et al. (2017) from the Williams Hills. This is a tentative interpretation because, if thickening is sea level controlled, there would be progressively less thinning expected upstream.

Our LGM ice thickness constraints are consistent with evidence for significantly thicker ice at the LGM in the Ellsworth Mountains (Hein et al., 2016, Fig. 2), and also likely consistent with measurements in Bentley et al. (2006). The post-LGM exposure ages of Hein et al. (2016) constrain LGM thicknesses to between 475, 373 and 247 m larger than present at three study sites in the Ellsworth Mountains. A pulse of up to 410 m of thinning appears similar both in scale and timing to the rapid ice surface lowering of 385 m recorded at the Lassiter Coast. Furthermore, measurements of long-lived nuclides by Bentley et al. (2006) show that there has been at least 300 m of thinning since the LGM in the Behrendt Mountains.

### 4.3 Grounding line position and flowline modelling comparison

Whitehouse et al. (2017) use their flowline model to reproduce the modern FIS ice surface profile and investigate the response of the ice stream to the onset of glacial and interglacial conditions. The following results from Whitehouse et al. (2017) are from their experiments in which the FIS is routed to the east of Berkner Island, which it is believed to have done during the LGM based on modelling studies (Le Brocq et al., 2011; Whitehouse et al., 2012) and aforementioned marine geological evidence for the former presence of grounded ice (Sect. 1.1). Under glacial conditions the FIS thickens by ~300 to ~500 m adjacent to the Thomas Hills, ~200 to ~400 m adjacent to the Williams Hills, ~150 to ~350 m adjacent to the Schmidt Hills, and ~100 to ~300 m proximal to the Shackleton Range. The lower value for each location is sourced from flowline experiments during which the grounding line of the FIS reaches a stable position at the northern margin of Berkner Island, with the higher value sourced from a scenario during which the grounded ice stream stabilises at the shelf edge. Our in situ [14]C LGM thickness constraints at each study location in the Pensacola Mountains and Shackleton Range exceed the upper estimates of the FIS flowline model of Whitehouse et al. (2017) under glacial conditions. The flowline model shows that the FIS, a major contributor to the total WSE ice flux, is able to reach a stable position at the shelf edge when tuned

using LGM thickness constraints lower than those presented here. Therefore, our thickness estimates add strength to the hypothesis that grounded ice occupying the WSE during the LGM reached a stable position located at the shelf edge (Bentley and Anderson, 1998; Hillenbrand et al., 2014).

### 4.4 Sea level contribution

To estimate the contribution to post-LGM sea level rise of the WSE we use a highly simplified scenario in which a range of minimum LGM thickness change estimates are distributed evenly across the WSE using an area for the sector defined by Hillenbrand et al. (2014). Distributing the lowest of our minimum LGM thickness constraints, 310 m for the Slessor Glacier, over the entire WSE produces a minimum sea level equivalent (SLE) of 2.2 m. When using the highest of our minimum thickness estimates, 800 m for the FIS, the minimum SLE increases to 5.8 m. Using the average minimum
LGM thickness constraint for our three study sites (580 m) produces a minimum SLE for the sector of 4.2 m. This scenario lacks any glaciological basis and is unrealistic, with no variation in ice thickness with location and no consideration of ice dynamics, isostasy, or bathymetry. Hence, further work is required to produce a realistic SLE for the WSE using our in situ [14]C thickness constraints.

      We compare our in situ [14]C LGM thickness estimates with the predicted LGM thickness change of three published
ice sheet models at each of our study sites to evaluate our minimum SLE estimates. We also quantify the WSE-sourced SLE for each model output. By comparing our data with the predicted LGM thickness from the model outputs, we can see which models predict LGM thickness changes in excess of and below our in situ [14]C thickness constraints. We compare our data with the predicted LGM thickness change at each of our sites from the ice sheet modelling of Le Brocq et al. (2011), Whitehouse et al. (2012), and Golledge et al. (2014), which predict a SLE for the WSE of ca. 3.0 m, 1.5 m, and 4.6 m,
respectively.

      From Fig. 12 it is apparent that the model output of Golledge et al. (2014) exceeds the thickness constraints at each of our sites. With a SLE of ca. 4.6 m for the WSE, this places a more robust upper limit on the minimum SLE contribution of the WSE using our data, showing that our upper minimum SLE estimate of 5.8 m is likely an overestimation due to the limitations outlined above. The only site where our minimum LGM ice thickness constraint exceeds any of the predicted
LGM thickness changes from the model outputs is at the Lassiter Coast, where a LGM thickness of 385 m larger than present exceeds the model output-based thickness estimate of both Le Brocq et al. (2011) and Whitehouse et al. (2012). The Lassiter Coast data indicate that the lower limit for the SLE for the WSE is between 3.0 m and 4.6 m, whilst evidence from all other sites suggests it was <1.5 m.

      Based on the above, we conclude that our minimum LGM thickness constraints indicate that the WSE contributed
<4.6 m, and possibly as little as <1.5 m, toward postglacial sea level rise. This is a range of minimum contributions to sea level rise and not a minimum-maximum range, as the values are informed using only minimum thickness constraints. Because this is an estimate for the lower limit of the SLE for the WSE, we cannot rule out a larger contribution.

      A SLE value of <4.6 m places our estimate between those modelled by Bentley et al. (2010) (1.4 m to 2 m) and Bassett et al. (2007) (13.1 to 14.1 m). Using the estimate based on all sites with the exception of the Lassiter Coast data, the
minimum SLE estimate of <1.5 m is consistent with the lower end of published SLEs for the sector. Our exposure ages indicate the Weddell Sea sector contributed to sea level during the early- to mid-Holocene, though they do not preclude a significant contribution earlier than this. Our estimates imply that the sector provided a modest contribution to global sea level. Whitehouse et al. (2017) estimate the sea level contribution of the FIS to between ~0.05 and ~0.13 m. Given that our [14]C thickness constraints for the FIS, including those in the Shackleton Range, exceed all of those used by Whitehouse et al.
(2017) to tune their flowline model, we propose that the sea level contribution for the FIS was greater than their upper estimate of ~0.13 m.

## 5. Conclusions

      We present LGM ice thickness constraints for three locations within the WSE of Antarctica. In situ [14]C measurements constrain the LGM thickness of the Foundation Ice Stream to at least ca. 800 m thicker than present in the
Schmidt Hills and at least 320 m thicker than present in the Thomas Hills, both in the Pensacola Mountains. The Slessor

Glacier was at least 310 m and up to 655 m thicker than present at the LGM. Finally, LGM ice was at least 385 m thicker than present at the Lassiter Coast. Our thickness constraints resolve a significant disconnect between previous terrestrial evidence for minimal LGM thickening in some locations from long-lived nuclides, and marine evidence for a significantly laterally expanded ice sheet with the grounding line located at the shelf edge. Our in situ $^{14}$C measurements made from samples at the Schmidt Hills exhibit higher than expected scatter in replicate measurements. Identifying the source of excess scatter will take further work. In terms of the contribution of the ice sheet sector to global sea level rise since the LGM, we estimate, primarily based on minimum estimates which do not constrain the upper limit of ice thickness changes, that the WSE contributed <4.6 m, and possibly <1.5 m.

**Data availability**

All sample data, including photographs when available, are available in the Informal Cosmogenic-Nuclide Exposure-Age Database (ICE-D) (http://antarctica.ice-d.org).

**Author contributions**

The study was conceived by BG and GB. Sample material was originally collected and provided by AH, JJ, GB, and CT. Sample preparation and analysis undertaken by KN and BG. Manuscript was written by KN with help from all authors.

**Competing interests:**

The authors declare that they have no conflict of interest.

**Acknowledgements**

KN and BG acknowledge support from NSF-OPP grant 1542936. GB acknowledges support from NSF-OPP 1542976 and from the Ann and Gordon Getty Foundation. Geospatial support for this work provided by the Polar Geospatial Center under NSF-OPP awards 1043681 and 1559691. This work forms part of the British Antarctic Survey 'Polar Science for Planet Earth' programme, funded by the Natural Environment Research Council.

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

# Figures

**Figure 1:** The Weddell Sea Embayment, including all locations referred to within the text. SH, WH and TH are the Schmidt, Williams and Thomas Hills, respectively. FH, P/M and MH are the Flower Hills, Patriot and Marble Hills, and the Meyer Hills, respectively. Black is exposed rock. Red boxes show extent of satellite images in Fig. 4. Exposed rock and coastline sourced from the SCAR Antarctic Digital Database. Bathymetry sourced from the International Bathymetric Chart of the Southern Ocean V1.0 (IBSCO; Arndt et al., 2013). Surface topography (shading) is sourced from the Reference Elevation Model of Antarctica (REMA; Howat et al., 2019). PS1423-2 is a marine sediment core from Crawford et al. (1996).

**Figure 2:** Current terrestrial ice thickness constraints inferred from measurements of long-lived nuclides around the WSE. Acronyms are as in Fig. 1. Constraints for the SH, WH, and TH are sourced from Balco et al. (2016) and Bentley et al. (2017). MB is Mount Bragg (Bentley et al., 2017). Thickness estimate for the Dufek Massif (DM) is sourced from Hodgson et al. (2012). Constraints for the P/M are sourced from Hein et al. (2016). For the MH and FH, the LGM thickness constraints are sourced from Fogwill et al. (2014). The thickness constraints sourced from Fogwill et al. (2014) were interpreted using modern ice surface elevations for the Rutford Ice Stream and Union Glacier measured using the Reference Elevation Model of Antarctica (REMA; Howat et al., 2019). Thickness constraints for the Shackleton Range are sourced from Hein et al. (2011, 2014). The range of LGM thicknesses for the Behrendt Mountains are sourced from multiple locations (Bentley et al., 2006).

**Figure 3:** Left: Hypothetical ice surface elevation change at a nunatak partially covered by ice at the LGM. Right: Resulting in situ $^{14}$C concentration, assuming no surface erosion, in samples collected at 100 m intervals along an elevation transect on the surface of the nunatak. Thin black lines indicate isochrons of exposure duration. Thick black line with dashed lines either side represent the saturation concentration and associated error envelope. Error envelope represents typical analytical uncertainty. "True exposure" refers to the resulting $^{14}$C concentration associated with the ice surface change history on the left plot. "Apparent exposure" is the resulting concentration that includes an inherited component, which is a residual $^{14}$C inventory remaining from the hypothetical samples which were saturated prior to ice cover.

**Figure 4:** Landsat imagery of study sites. Location of each image is shown in Fig. 1. Green dots show sample locations. Arrows show ice flow directions. A: Mt. Skidmore and Mt. Provender, Shackleton Range. B: Lassiter Coast, southern Palmer Land. C: Schmidt Hills, Pensacola Mountains. D: Thomas Hills, Pensacola Mountains. Landsat 8 imagery courtesy of the U.S. Geological Survey. Grounding line positions sourced from the MEaSUREs program V2 (Rignot et al., 2011, 2014, 2016).

**Figure 5:** Left: Elevation versus in situ $^{14}$C concentration of samples from the Shackleton Range. Circles are erratic cobbles, triangles are bedrock. Some error bars are smaller than their respective data points. Horizontal dashed lines show the approximate elevation of the modern ice surface at each site. Light grey lines indicate isochrons of exposure duration. Thick black line and grey shading are the saturation concentration and associated error envelope. Right: Exposure ages from this study (in situ $^{14}$C) and $^{10}$Be ages of Hein et al. (2011, 2014). Samples yielding infinite in situ $^{14}$C ages are not presented on the right-hand plot.

**Figure 6:** Left: Elevation versus in situ $^{14}$C concentration of samples collected from the Lassiter Coast. All samples are bedrock. Right: In situ $^{14}$C exposure ages with $^{10}$Be ages of Johnson et al. (2019).

**Figure 7:** Left: Elevation versus in situ $^{14}$C concentration of samples collected from the Schmidt Hills. All samples are erratics with the exception of the highest elevation sample, shown with a triangle. Samples with replicate measurements are displayed with differing symbols. Right: Schmidt Hills exposure ages from this study (in situ $^{14}$C) and those of Balco et al. (2016) and Bentley et al. (2017) ($^{10}$Be). Measurements yielding infinite in situ $^{14}$C ages are not presented on the right-hand plot.

**Figure 8:** Left: Elevation versus in situ $^{14}$C concentration of samples collected from the Thomas Hills. All samples are erratics. Note that both plots contain in situ $^{14}$C data for two samples within close agreement, such that the points overlap. Right: Thomas Hills exposure ages from this study (in situ $^{14}$C) and those of Balco et al. (2016) and Bentley et al. (2017) ($^{10}$Be).

**Figure 9:** Terrestrial ice thickness constraints inferred from measurements of cosmogenic nuclides around the WSE. Constraints for the Lassiter Coast (LC), Shackleton Range, and the Schmidt Hills are sourced from this study. All other ice thickness values and locations are the same as in Fig. 2.

**Figure 10:** Exposure-age results projected onto an elevation profile along flowline of the Slessor Glacier. Flowline location is shown in the map (right). Infinite [14]C measurements are offset in regard to their distance along flowline to improve readability. The [10]Be data included are those from Hein et al. (2011, 2014) which yield exposure ages below 12 ka (LSDn scaling, antarctica.ice-d.org). Elevation data for ice surfaces and map shading is sourced from the Reference Elevation Model of Antarctica (REMA; Howat et al., 2018). Grounding line positions sourced from the MEaSUREs program V2 (Rignot et al., 2011, 2014, 2016). Minimum LGM surface is the modern day surface profile with the elevation increased above present using our minimum LGM thickness estimates.

**Figure 11:** Exposure-age results projected onto an elevation profile along flowline of the Foundation Ice Stream. Flowline location is shown in the map (right). Infinite [14]C measurements are offset in regard to their distance along flowline to improve readability. The [10]Be data included are those from Balco et al. (2016) and Bentley et al. (2017) which yield exposure ages below 12 ka (LSDn scaling, antarctica.ice-d.org). Elevation data for ice surfaces and map shading is sourced from the Reference Elevation Model of Antarctica (REMA; Howat et al., 2019). Local ice margins are highly simplified. Grounding line positions are sourced from the MEaSUREs program V2 (Rignot et al., 2011, 2014, 2016)). Minimum LGM surface is the modern day surface profile with the elevation increased above present using our minimum LGM thickness estimates.

**Figure 12:** Predicted LGM ice thickness change from three ice sheet model outputs at each of our study sites and their associated sea level equivalent (SLE) for the Weddell Sea Embayment (WSE) sector of the Antarctic ice sheet. A. is Mt. Skidmore, B. Mt. Provender, C. the Lassiter Coast, D. Schmidt Hills, and E. Thomas Hills. Vertical blue lines show the interpreted LGM thickness change at each site based on our in situ [14]C data. For A. and B., the two vertical blue lines show the range of thickness estimates for the two sites, with the upper limit constrained by the highest elevation saturated sample at Mt. Provender. "G2014" refers to Golledge et al. (2014), "LB2011" refers to Le Brocq et al. (2011), and "W2012" refers to Whitehouse et al. (2012). Errors are not provided for the model outputs. The average error of published SLEs associated with model outputs for the entire ice sheet is 1.45 m (see Simms et al., 2019). We therefore use an error of 0.3 m for the three model SLEs, which is 22% of the average error (22% is the proportion of the AIS that the WSE drains, see Joughin et al. (2006)).

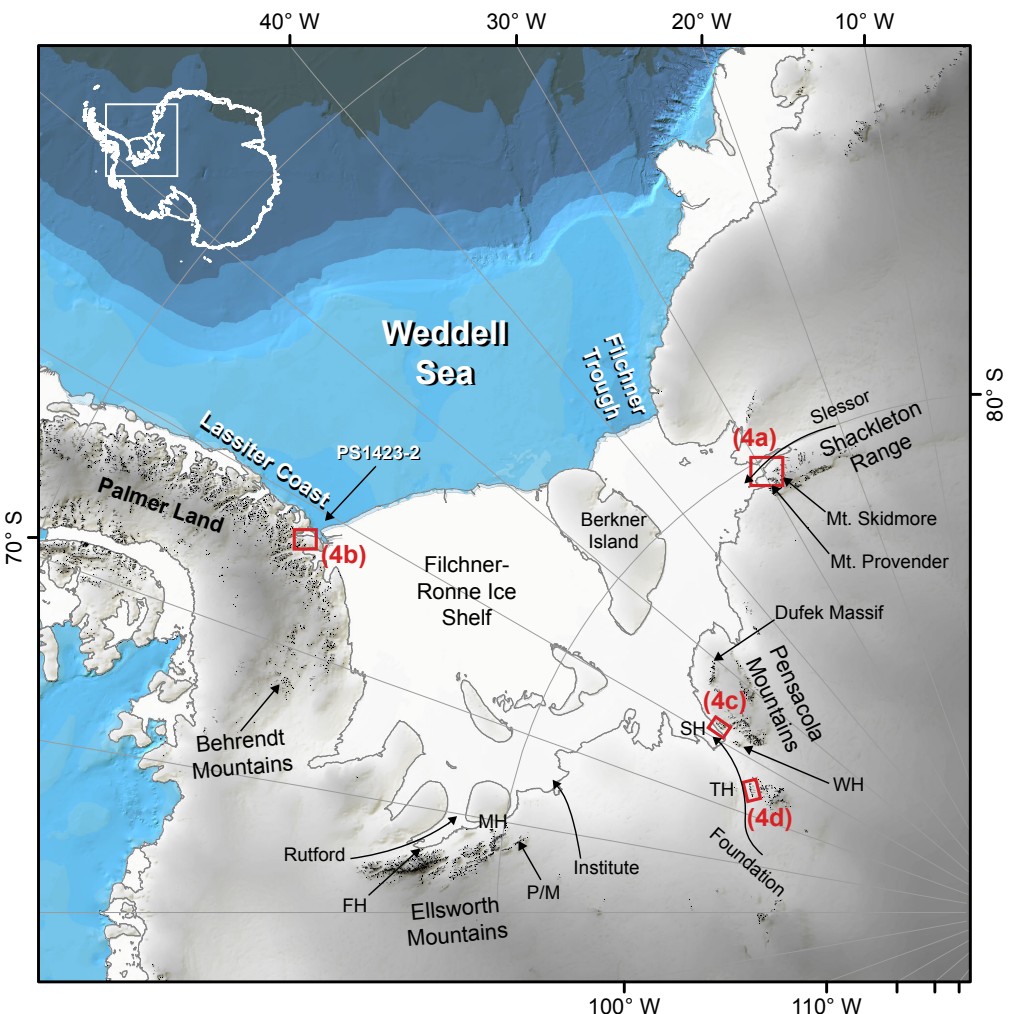

Fig. 1

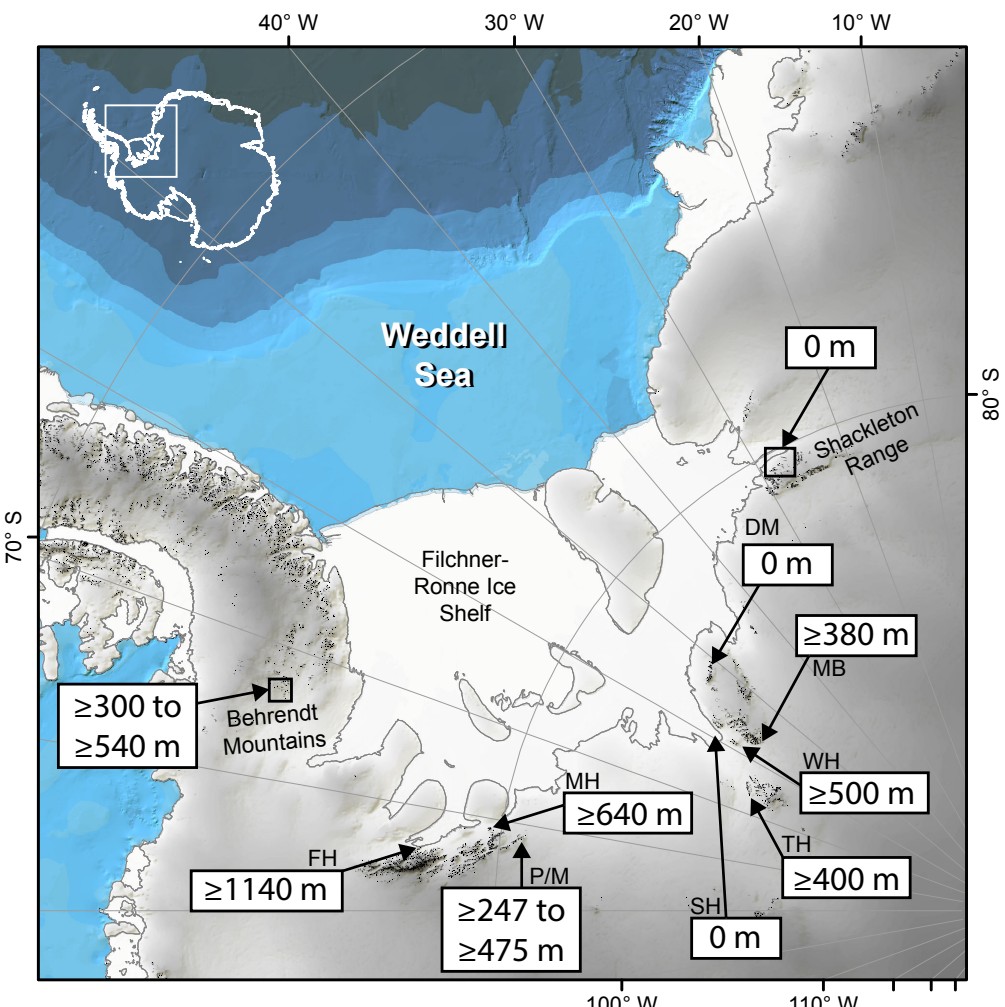

Fig. 2

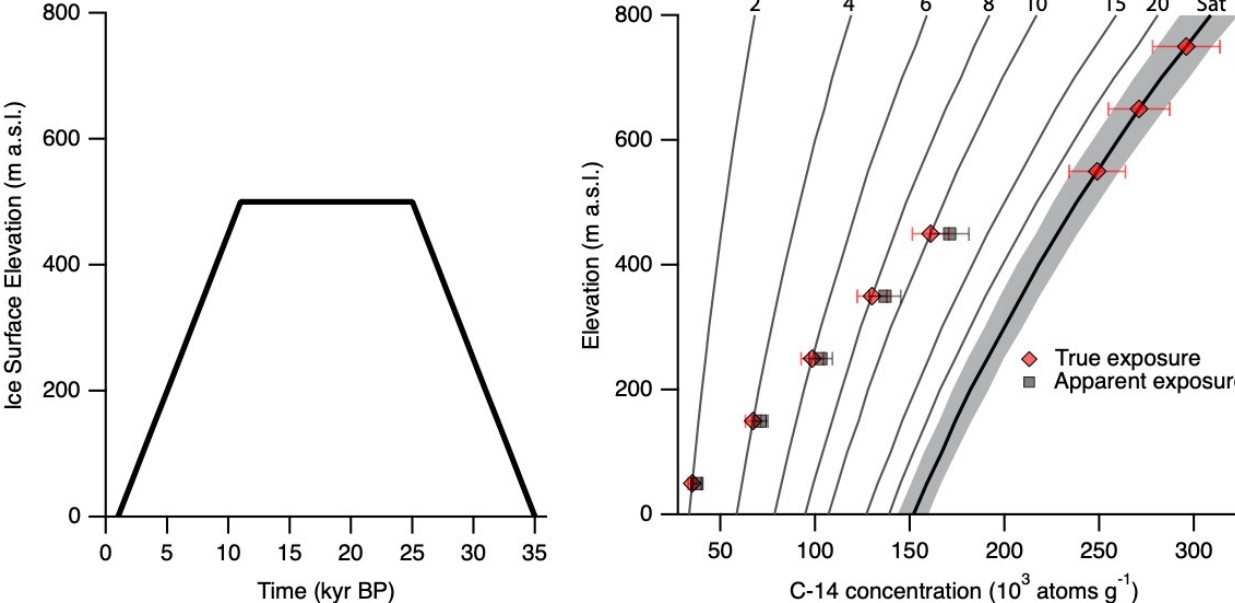

Fig. 3

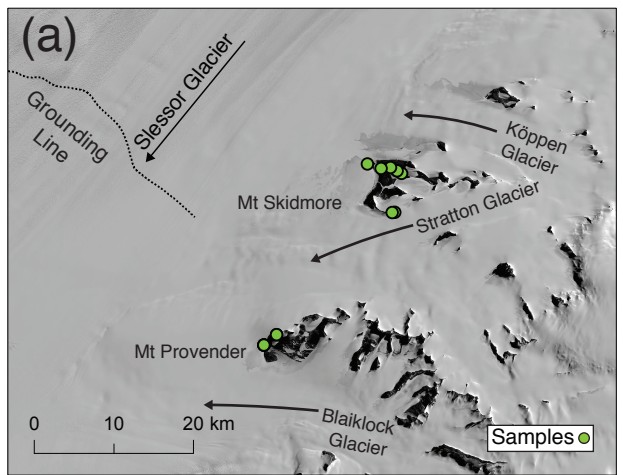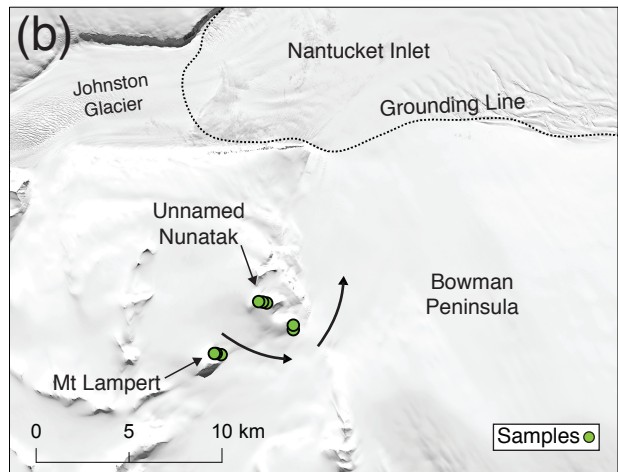
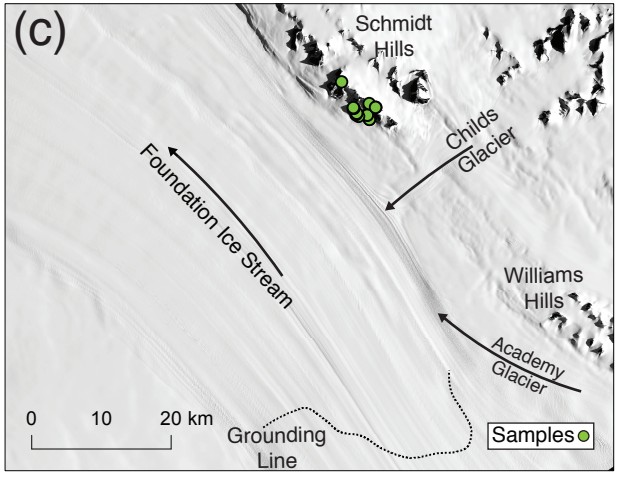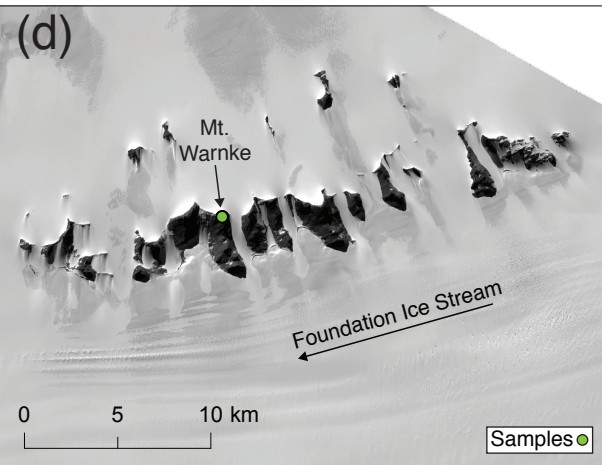

Fig. 4

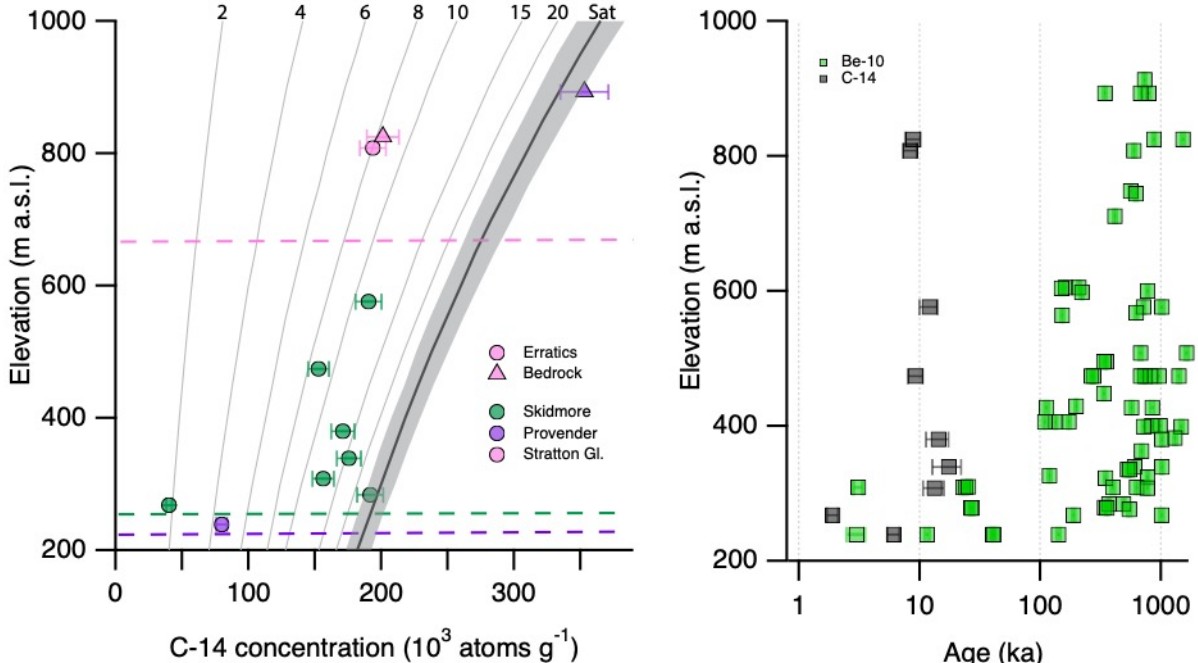

Fig. 5

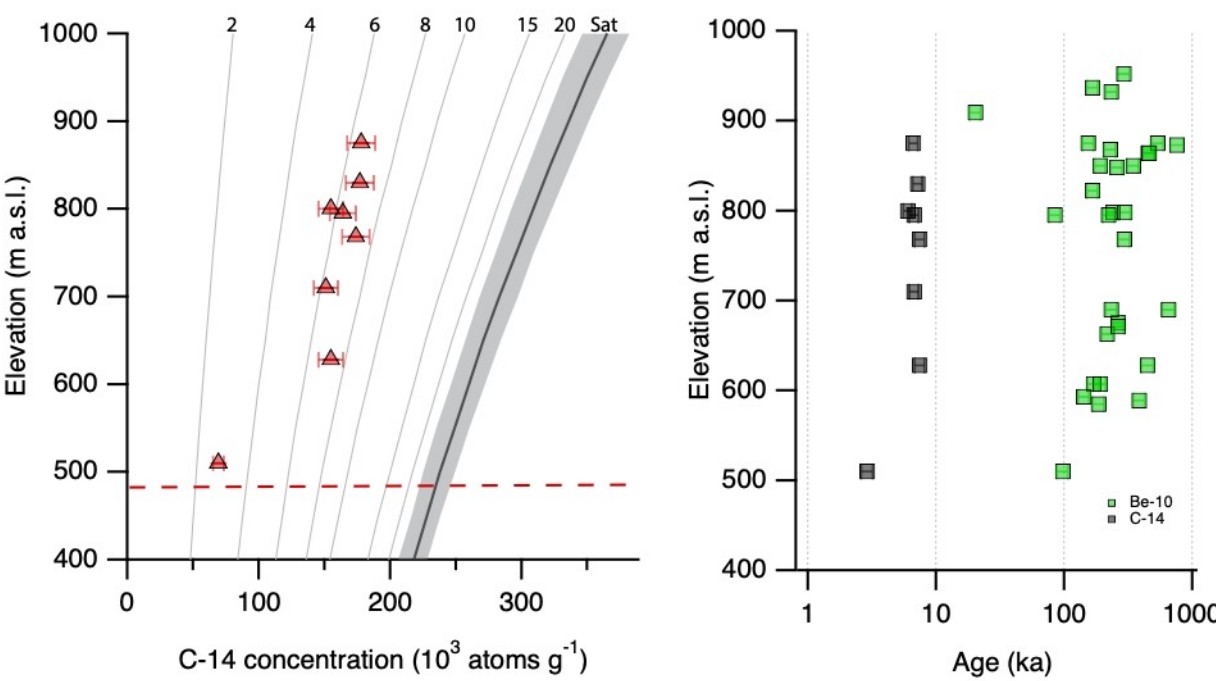

Fig. 6

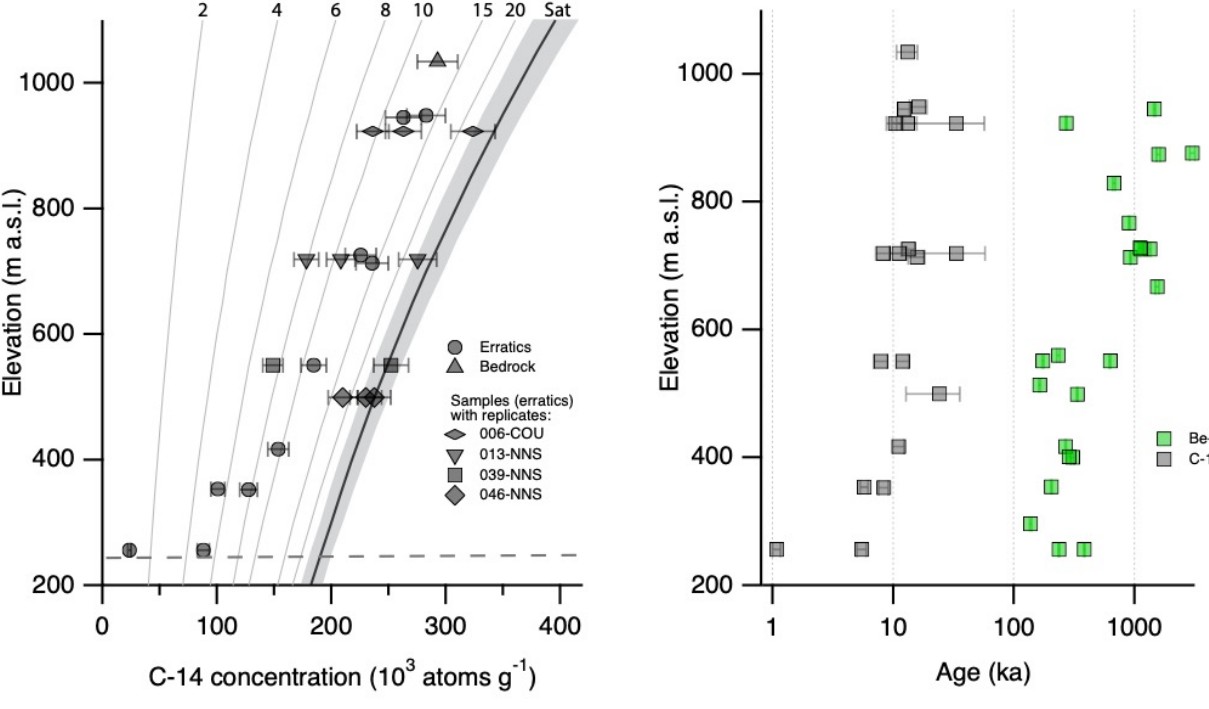

Fig. 7

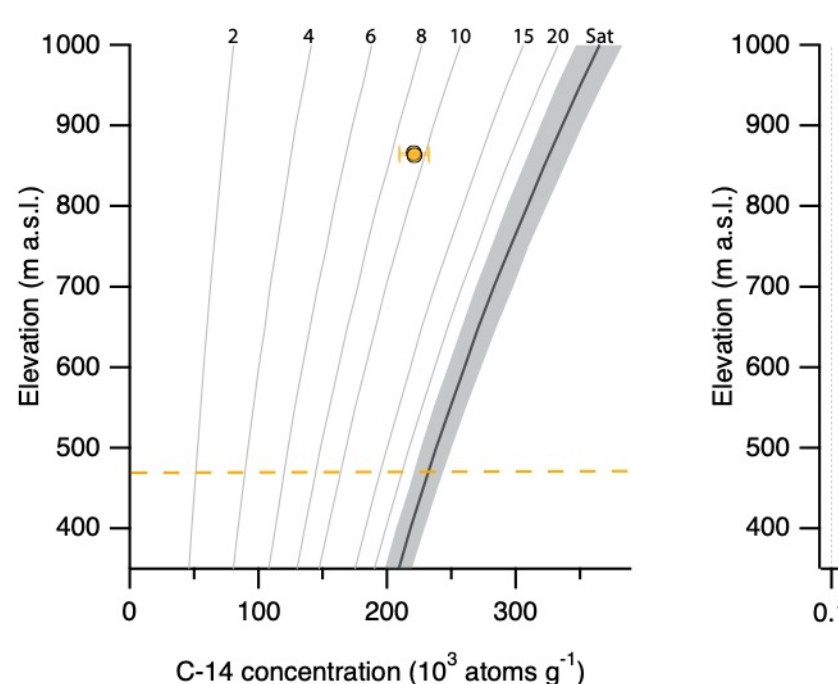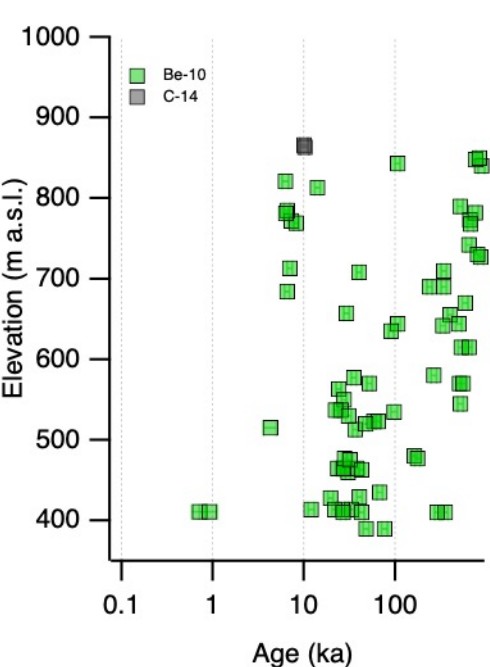

Fig. 8

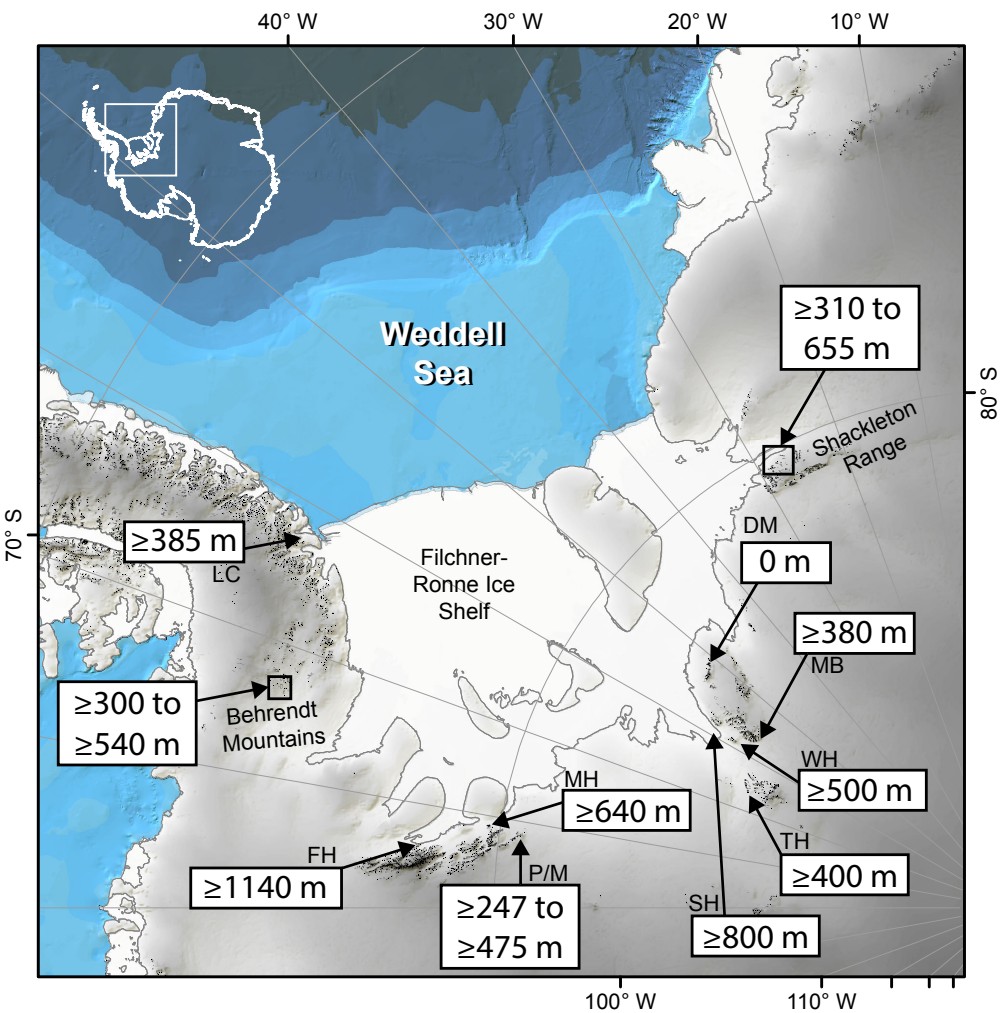

5    Fig. 9

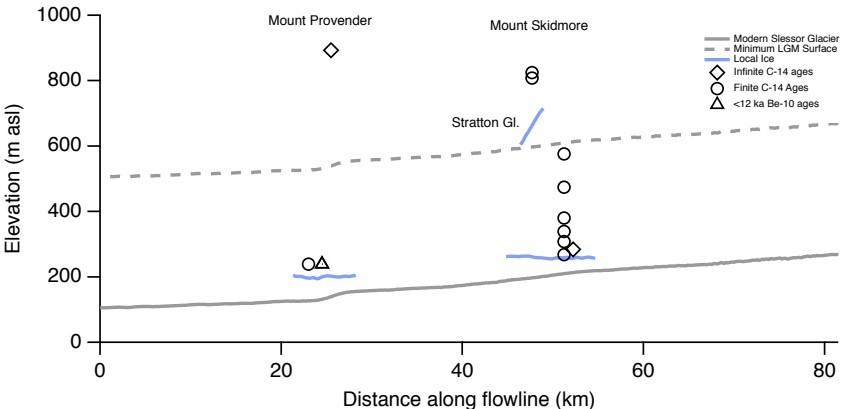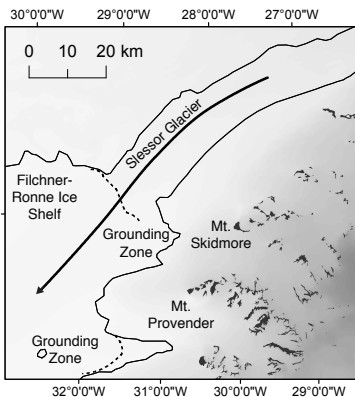

Fig. 10

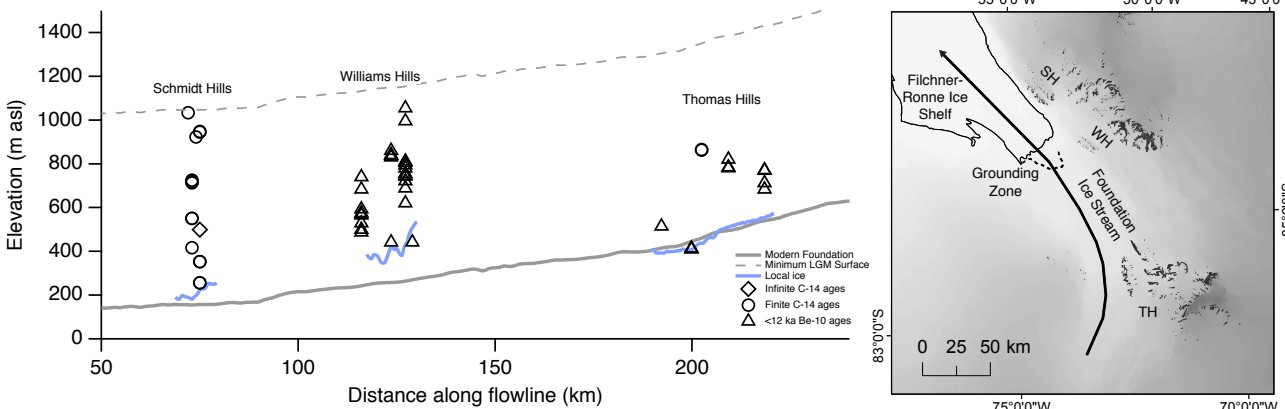

Fig. 11

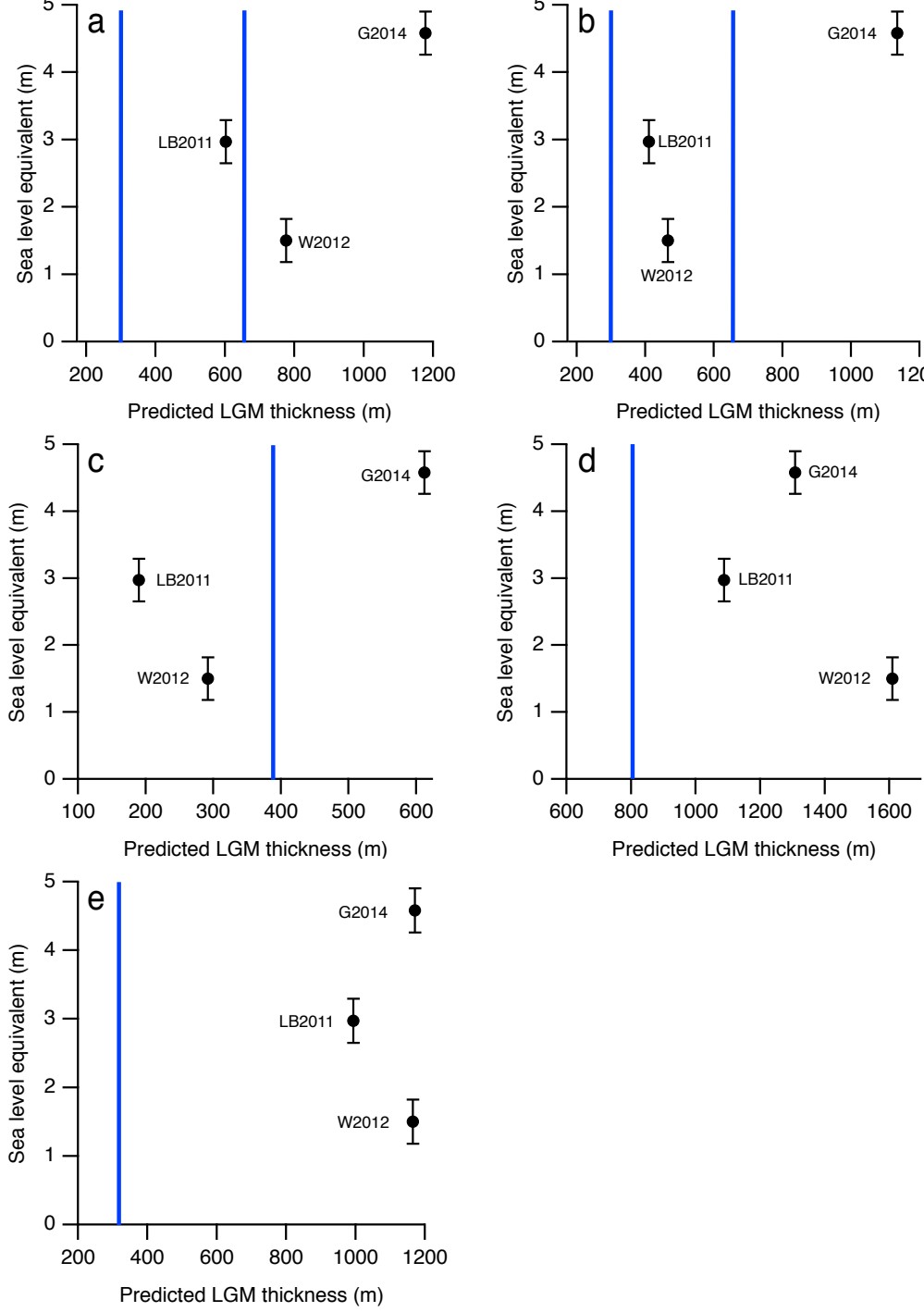

Fig. 12