# Peer review of "New Last Glacial Maximum Ice Thickness constraints for the Weddell Sea Embayment, Antarctica"

_The Cryosphere, 2019_

## Referee Comment (RC1) · Anonymous Referee #1 · 7 Jun 2019

Overview

This paper reports an in situ-produced 14C chronology of past ice thickness change for several sites around the Weddell Sea Embayment. Reconstructions of Last Glacial Maximum (LGM) ice thicknesses and subsequent deglaciation have been complicated onshore by a lack of reliable constraints, primarily due to issues with the cosmogenic inheritance of 10Be. Here the authors attempt to improve on existing LGM constraints from the region, and present an approach to determine whether sites were covered at the LGM using the concentration saturation point of in-situ 14C. Evidence is found for LGM thickening at all sites, with an upper limit provided at one site. The new constraints help resolve apparent differences between marine and terrestrial records. They conclude by providing sea-level equivalent estimates of ice volume from their new ice thickness constraints.

Main comments

The paper presents novel data to address the important topic of LGM ice volume. Notably, the ice sheet was at least 310-800 m thicker in the Weddell Sea Embayment. The results are definitely within the scope of The Cryosphere, and will be of interest to those reconstructing past ice sheet change using empirical data, ice sheet modellers and glacial isostatic adjustment modellers. I enjoyed reading the paper, which is well-written with clear figures. The title is suitable and the abstract concisely summarises the paper. The methods are described in adequate detail (the schematic in Fig. 1 is very useful), the treatment of the data seems valid, and the limitations of the technique and data are well-discussed.

My main issue with the paper is how the data are discussed in relation to the maximum LGM ice thickness and the contribution to deglacial sea-level rise (primarily sections 4.2 and 4.4). Firstly, the highest sample at Mt Provender produces a 14C concentration that appears to have reached saturation, which would imply ice cover at 30-35 kyr. But, as the authors discuss in detail for Mt Skidmore and Schmidt Hills, bogus high 14C concentrations can result from analytical and geomorphic issues. I would therefore be justified to argue that the single sample at Mt Provender is not sufficient to provide an upper LGM thickness limit, and that all of the data presented in the paper only provide minimum LGM ice thickness estimates.

Secondly, without maximum constraints on LGM ice thickness, it is not possible to infer when the majority of post-LGM ice thickness occurred. Specifically, the exposure ages do not indicate that the Weddell Sea sector contributed to sea-level most significantly during the early-mid Holocene (section 4.4, line22), and the majority of exposure ages occurring post-MWP1a does not mean that this sector didn't contribute significantly to

sea-level rise at this time as there is limited data constraining maximum ice thickness prior to MWP1a (4.4, lines 22-26). It also appears in Figures 5 (Mt Skidmore) and 7 (Schmidt Hills) that accelerated thinning may have occurred at about the time of MWP1a, but the logarithmic axis prevents the reader from assessing this.

Minor comments

Page 1, line 30: It is necessary to broadly define the time of the Last Glacial Maximum, and briefly discuss global vs regional differences, as this has implications for what you are trying to achieve.

Page 2, line 13: "Results from Schmidt Hills (Fig. 1)..." do you mean Fig. 2?

Page 3, line 17: You should state the half-lives of 10Be and 26Al.

Page 3, lines 19-20: Reduce concentrations to what level? To below analytical uncertainty?

Page 3, line 23: "before the LGM", as above, the time of the LGM has not been defined.

Page 3, lines 44-45: Erratics can be considered bedrock, but with the assumption that they have not been transported from upstream during periods of ice cover.

Page 4, lines 2-5: What about surface erosion? This should be acknowledged here.

Page 5, line 9: 320 m above the FIS ice margin, but are very close to ice at a much higher elevation on the other side of the range. Which ice surface likely covered the samples? Why?

Page 5, lines 34-35: How well known is it that the sampling location was not covered by ice at the LGM?

Page 6, lines 6-7: The "upper limit" is based on a single sample as discussed above. How reliable is this? This limitation should be acknowledged here and discussed in section 4.

[Figure]

Page 7, line 44: What is the statement that the entirety of Mt Skidmore was covered by ice at the LGM based on? How high is Mt Skidmore? Is there a post-LGM sample at the top?

Page 8, line 3: Again, only a single sample supports Mt Provender remaining exposed at the LGM.

Page 9, lines 1-4: It is probably overly precise to report estimates of modelled ice thickness to the nearest metre given the uncertainty in the modelling experiments.

Page 9, lines 16-17: "Using the average LGM thickness constraint" is a little misleading as it is the average of minimum ice thickness estimates, rather than an average of a suite of maximum and minimum estimates.

Page 9, lines 17-20: How would you expect these simple estimates to differ if you were to account for spatial variation in ice thickness, etc.? At the very least it is probably fair to say that there was less thickening inland relative to at the sample sites and the ice sheet margin.

Page 9, line 24: Unnecessary to abbreviate to MWP1A as only referred to one other time.

Page 9, line 35: Possibly "up to 655 m…".

In several places: Both ka and kyr is used. Pick one.

References: Fogwill et al. (2014), and maybe other papers, are missing.

Fig. 1: Should 2a, 2b, … not be labelled 4a, 4b, …?

Fig. 3: What is the error envelope based on? What uncertainty has been used for the hypothetical sample concentrations? Typical analytical uncertainty?

Supplementary material:

Would be useful if the study site figures had contours (e.g. from REMA).

Would be nice to see a figure with the 14C ages plotted on a linear axis, perhaps vs the relative elevation of the samples.

---

## Referee Comment (RC2) · Jane L. Andersen (Referee) · 7 Jun 2019

General comments: This paper presents new in-situ 14C ages (+ replicate measurements) to constrain the glaciation history of the Antarctic ice sheets in the Weddell Sea sector. Such data are highly necessary in order to tease out non-erosive burial of bedrock and erratics, which due to the often cold-based nature of the ice sheet margins outside of major troughs, is a prevalent problem for the interpretation of existing cosmogenic records using longer-lived nuclides. This study thus provides an important contribution that I expect will draw great interest from the scientific community. The paper is well-written, figures are good, and the data presented clearly. My main concerns

are as follows:

1. The replicate measurements demonstrate that there are serious problems with the reproducibility of this dataset. In the current form, this comes as a complete surprise halfway through the manuscript, as it is not reported in the abstract, introduction or conclusion were the results are stated without mentioning this important issue. I think this issue should be made more transparent throughout the manuscript, and that the implications for the non-replicated data should be discussed in more detail. Given the authors' argument that the saturated samples are more prone to being erroneous due to contamination issues with modern carbon, I am specifically concerned with whether too much emphasis is put on the single saturated sample from Mount Provender. Is there any evidence that this sample should be more reliably saturated than the replicated ones? This is particularly important since this is the only data point within the dataset that could otherwise provide a maximum ice sheet thickness during LGM. Furthermore, since this issue challenges the premise of the paper - that the maximum LGM thickness can be detected by the limit between saturated and unsaturated samples in an in situ 14C-elevation profile - I would like to see a brief discussion on how this issue could be handled in the future. Since measurements have been done on samples collected by others with the aim of dating with long-lived nuclides, such a discussion could further include a description of a sampling strategy that could better test your basic premise, would you e.g. recommend sampling bedrock for future applications of this method?

2. I find the presentation and discussion of the implication of these data for the Weddell sea sector's contributions to LGM-present sea level somewhat misleading, and think the authors should give this subject some more consideration. Firstly, I think the estimated range of 2.2-5.8 m is somewhat deceiving as it gives the impression that this is a minimum-maximum envelope, while it is really based on minima estimates from two different locations. It should be clearly stated that these are both minimum estimates. Perhaps it would be more appropriate to make a single minimum estimate based on

e.g. a smoothed surface fit to the minima-elevations. Secondly, I don't see how the author's interpretation of a limited MWP-1A contribution from this sector of the AIS is based on the available cosmogenic data. In my opinion, the results provide robust minimum estimates of the maximum thickness during LGM, and Holocene thinning rates at sites where (sampled) nunatak elevations coincide with ice sheet surface lowering. Yet, given that the only maximum ice thickness estimate is based on a single saturated sample (which again is less certain owing to the replication issue mentioned above), nothing can be reliably said about thinning rates prior to the early Holocene.

Specific comments:

P1, L20-22: specify already here that you use a 'short-lived' cosmogenic nuclide, which is less susceptible to inheritance problems than 10Be and other long-live nuclides.

P2, L21: 'without erosion beneath cold-based ice' is confusing, perhaps 'when protected from erosion beneath cold-based ice'.

P2, L22: 'reduce concentrations' - concentrations will start to reduce immediately by decay when buried, specify 'to below measurable levels' or similar.

P2, L24-25: briefly specify somewhere around this transition in the paper that in-situ 14C is less sensitive to inheritance due to its short half-live, this is key to understand the difference between using one nuclide or another. Also state half-lives of 10Be (+ 26Al) here or elsewhere.

P2, L25&27: you are not really discussing and constraining 'LGM thickening', but the maximum thickness and subsequent thinning.

P3, L19: again, 'reduce' on its own is confusing, add 'to below analytical limit' or similar. Also specify what you mean by 'short' by stating the approximate burial time it would typically require.

P3, L21: 'surfaces not covered by ice during the LGM', this could be stated more generally, e.g. 'continuously-exposed, slow-eroding surfaces'
P3, L22: it would also require low erosion rates to reach such high apparent ages, specify this.

P3, L25: cold-based ice has been shown to sometimes erode (e.g. Atkins et al., 2002, Geology v30 p.659-662). Perhaps specify 'cold-based, non-erosive ice' or simply 'non-erosive ice'

P3, L38 – P4, L4: I think the discussion of whether the erratics record the history of their sampling location could be made more concise. It shouldn't matter for in-situ 14C whether the erratic has been 'covered and uncovered' in place or during transport, as long as it has been buried long enough and then deposited by the ice, with no subsequent movement? Also avoid moving back-and-forth between discussing bedrock and erratics. P3, L46: specify which assumption.

P4, L9-11: I don't think it is necessary to repeat this information here

P6, L. 43-46: At present this section has a bit of an 'ad-hoc argument' feel to it, I think it would be better to first provide an objective discussion on what criteria can be used to assess the validity of this method before you go through each site.

P6, L. 44: I don't think the Lassiter coast dataset can be strictly described as having a 'linear age-elevation relation', as it requires at least two lines two fit the data. Perhaps rephrase to 'ages continuously decrease towards the present ice sheet surface'.

P7, L7: 008-NNS doesn't seem to be replicated here? I don't find a saturated (replica-) measurement in the table or figure.

P7, L12-13: You don't present replicate measurements from elsewhere than the Schmidt hills, so this sentence reads a bit strange.

P7, L14: This transition was not obvious to me. This also goes for the next two paragraphs, I think you could link the various paragraphs discussing the poor replicability better. The 'Regardless,' makes it sounds like you are moving on to another subject. This paragraph is discussing whether there could be any geological reasons that the

in-situ 14C profiles are 'inverted'?

P7, L22: specify 'erratic' samples

P7, L26-27: I couldn't follow this argument.

P7, L37: I think you need to discuss here whether or not you expect the same issue for the saturated, but non-replicated sample from Mount Provender. Given only two data points from this site, the fact that the 'profile' isn't inverted is not a very strong argument, or?

P8, L11: provide the in situ 14C ages with uncertainty here to make it easy for the reader to compare

P9, L23: You first state that your data 'do not preclude a significant contribution earlier than (the early to mid-Holocene)', yet move directly on to suggest that MWP1A was not significant in this region, thus contra-arguing yourself. I don't think you can constrain MWP1A thinning on the basis of these data.

Fig. 1: boxes should refer to Fig 4a-d, not 2a-d.

Fig. 2: legend P13, L13: remove 'constraints' before 'ice'

Fig. 3: specify that you used zero erosion (presumably) for these calculations. It looks strange to me that the saturated samples are labelled as 'true' exposure. The figure is a bit blurry compared to the others.

Fig. 4: green color of sample points in legend and on map does not appear to be the same

Fig. 5: add legend showing circle=erratic, triangle=bedrock.

Fig. 9+10: legend refers to maps on left, but they are on the right

Technical corrections:

P1, L32: Should Antarctic ice sheet(s) be in plural, west+east?

P2, L18: specify 'non-erosive' burial

P3, L22: 'measured today' seems unnecessary

P4, L31: remove 'from'

P5, L13: 'analysed' and 'analysis' in same short sentence

P6, L18: perhaps 'ice thinning' rather than 'deglaciation'?

P7, L17: fig 10 referred before fig. 9. Fontsize is odd

P10, L6-7: BG rather than BM?

Supplementary table S2: Spreadsheet tab is named 'Table 3' while table is 'Table 2'.

---

## Author Comment (AC1) · 2 Aug 2019

We would like to thank reviewer 1 for the thorough, thoughtful and constructive review.

Text in bold and italics are comments made by the reviewer, with responses in regular text following. A revised manuscript with tracked changes made following comments from both reviewers is included following our responses.

*Reviewer 1 Main comments*

***My main issue with the paper is how the data are discussed in relation to the maximum LGM ice thickness and the contribution to deglacial sea-level rise (primarily sections 4.2 and 4.4). Firstly, the highest sample at Mt Provender produces a 14C concentration that appears to have reached saturation, which would imply ice cover at 30-35 kyr. But, as the authors discuss in detail for Mt Skidmore and Schmidt Hills, bogus high 14C concentrations can result from analytical and geomorphic issues. I would therefore be justified to argue that the single sample at Mt Provender is not sufficient to provide an upper LGM thickness limit, and that all of the data presented in the paper only provide minimum LGM ice thickness estimates.***

The saturation concentration of the sample in question indicates that it has remained exposed since 30 to 35 ka, but we don't know if that means it was covered by ice at 30 to 35 ka or prior to that. The sample tells us that there has been a lack of ice cover for 30 to 35 kyr but doesn't imply ice cover at any time. We have altered the text to highlight the limitation that the maximum ice thickness change for the Slessor Glacier is based on one sample. We now state that "Our measurements show that the Slessor Glacier was between 310 and up to 655 m" (P1, line 24, see also P10, line 1 in Conclusion). We also highlight the limitation in the Results (P6, lines 22 to 23) and the Discussion.

In the Discussion (P8, lines 37 to 41), we state the following:

"Whilst the upper limit of LGM ice at Mt. Provender is based on a single sample, we believe this sample is a reliable indicator of LGM ice thickness for the following reasons. The sample is sourced from bedrock and therefore cannot have been subjected to geomorphic scenarios causing the exposure age to misrepresent the timing of ice retreat. Furthermore, froth flotation, which introduces modern carbon to sample material (Nichols and Goehring, in review), was not used to isolate quartz for this sample."

***Secondly, without maximum constraints on LGM ice thickness, it is not possible to infer when the majority of post-LGM ice thickness occurred. Specifically, the exposure ages do not indicate that the Weddell Sea sector contributed to sea-level most significantly during the early-mid Holocene (section 4.4, line22), and the majority of exposure ages occurring post-MWP1a does not mean that this sector didn't contribute significantly to sea-level rise at this time as there is limited data constraining maximum ice thickness prior to MWP1a (4.4, lines 22-26). It also appears in Figures 5 (Mt Skidmore) and 7 (Schmidt Hills) that accelerated thinning may have occurred at about the time of MWP1a, but the logarithmic axis prevents the reader from assessing this.***

We agree with the comments from both reviewers on the inability of our data to provide information on the contribution of the sector the MWP1A, and when exactly the majority of post-LGM thinning occurred. We have thus removed any mention of MWP1A and any interpretation of when the majority of post-LGM thinning occurred.

***Minor comments***

***Page 1, line 30: It is necessary to broadly define the time of the Last Glacial Maximum, and briefly discuss global vs regional differences, as this has implications for what you***

*are trying to achieve.*

We have added a broad definition of the LGM (P1, lines 33-35). "We broadly define the LGM as the period between ~15 and 25 ka when the Antarctic ice sheet volume was near its maximum extent in the last glacial-interglacial cycle".

*Page 2, line 13: "Results from Schmidt Hills (Fig. 1): : :" do you mean Fig. 2?*

We have altered the text to fix this mistake.

*Page 3, line 17: You should state the half-lives of 10Be and 26Al.*

Added to P1, line 38.

*Page 3, lines 19-20: Reduce concentrations to what level? To below analytical uncertainty?*

We have added "to below analytical uncertainty" to the sentence in question.

*Page 3, line 23: "before the LGM", as above, the time of the LGM has not been defined.*

Added definition of LGM, see above response.

*Page 3, lines 44-45: Erratics can be considered bedrock, but with the assumption that they have not been transported from upstream during periods of ice cover.*

We think this is adequately covered by the preceding sentence "It is highly likely that these erratic samples have been repeatedly covered and exposed by cold-based ice." Being repeatedly covered and exposed with cold-based ice would require them to have been in place prior to the most recent ice advance, meaning that they have been covered repeatedly and built up hundreds of thousands to millions of years of $^{10}$Be (see previously published $^{10}$Be measurements in Figures 5 to 7 which are consistent with this assumption). There is also a general trend of decreasing $^{10}$Be concentration with decreasing elevation at all sites (erratics at the Schmidt Hills and Shackleton Range, bedrock at the Lassiter Coast), which may reflect the thinning and thickening of cold-based, non-erosive ice over the long term.

Any ice advances beyond the most recent are irrelevant for in situ $^{14}$C, given the half-life of the nuclide. If the erratics were transported by an earlier ice advance prior to ca. 35 ka, the in situ 14C concentration would not reflect this.

*Page 4, lines 2-5: What about surface erosion? This should be acknowledged here.*

Added to P4, line 12.

*Page 5, line 9: 320 m above the FIS ice margin, but are very close to ice at a much higher elevation on the other side of the range. Which ice surface likely covered the samples? Why?*

The other side of the range is a somewhat steep cliff. With the large catchment of the FIS, we assume that the FIS expanded and covered the Thomas Hills. The pattern of glacial deposits, with active blue ice moraines facing the Thomas Hills, proximal younger drift and then older drift at higher elevations (see Fig. 7 of Balco et al. (2016)) supports the previous expansion and contraction of the FIS covering the Thomas Hills.

***Page 5, lines 34-35: How well known is it that the sampling location was not covered by ice at the LGM?***

We added the following to P5, lines 39 to 43 ", based on geological mapping and an ash chronology, the sampling location has remained ice-free since >11.3 Ma (Marchant et al., 1993). All reported in situ 14C measurements from CRONUS-A, made at multiple laboratories, yield concentrations equivalent to saturation based on other calibration data from elsewhere in the world (e.g. Jull et al., 2015; Fülöp et al., 2018; Goehring et al., 2019; Lamp et al., 2019). We use the CRONUS-A measurements to calibrate the 14C production rate to reduce scaling extrapolations."

***Page 6, lines 6-7: The "upper limit" is based on a single sample as discussed above. How reliable is this? This limitation should be acknowledged here and discussed in section 4.***

We have acknowledged and discussed the limitation as per the first main concern above. See our response including and following "We have also altered the text to highlight the limitation that the maximum ice thickness change for the Slessor Glacier is based on one sample…"

***Page 7, line 44: What is the statement that the entirety of Mt Skidmore was covered by ice at the LGM based on? How high is Mt Skidmore? Is there a post-LGM sample at the top?***

The highest elevation samples from Mt. Skidmore are those that are described as being most proximal to the Stratton Glacier, sampled from ca. 808 and 825 m asl. There is a small exposed surface between Ice Tongue A and Ice Tongue B of the Stratton Glacier with a peak that reaches up to ca. 850 masl (according to REMA). Given the evidence for the expansion of the Stratton and Slessor glaciers, we think there is sufficient evidence to assume that this peak was covered by either the larger ice masses, or the smaller local ice tongues, at the LGM. We've added more information to the text (P8, lines 33 to 35) to highlight this.

***Page 8, line 3: Again, only a single sample supports Mt Provender remaining exposed at the LGM.***

Agreed, see our response to the first main concern above.

***Page 9, lines 1-4: It is probably overly precise to report estimates of modelled ice thickness to the nearest metre given the uncertainty in the modelling experiments.***

Agreed, we have changed the values to more fairly represent the uncertainty in the modelling experiments (P9, lines 39 to 41).

***Page 9, lines 16-17: "Using the average LGM thickness constraint" is a little misleading as it is the average of minimum ice thickness estimates, rather than an average of a suite of maximum and minimum estimates.***

Agreed, we have altered the methodology for Sect. 4.4 (P10, lines 5 to 35). We have added "minimum" whenever describing the LGM thickness constraints and resulting SLE estimates. Furthermore, we have added to our original approach to produce a sea-level contribution estimate that is more quantitative than those presented in the initial submission. We now compare the modelled LGM ice thickness at each site from a range of ice sheet model outputs to our in situ $^{14}$C-derived ice thickness interpretations. We look at the sea-level equivalent for the Weddell sector for each ice sheet model output used. We have used the ice sheet model outputs of Le Brocq et al. (2011), Whitehouse et al. (2012) and Golledge et al. (2014). The LGM thickness constraints at all of our study sites are smaller than the predicted LGM thickness change of Golledge et al. (2014), which predicts a sea-level equivalent of ca. 4.6 m for the Weddell sector, placing an upper limit on our minimum SLE equivalent estimate. We also make it clear in the text that we provide an estimate for the minimum SLE for the WSE, and we now only use the minimum LGM ice thickness estimates for the sea-level contribution discussion. We have also added a new figure (12) to this section.

***Page 9, lines 17-20: How would you expect these simple estimates to differ if you were to account for spatial variation in ice thickness, etc.? At the very least it is probably fair to say that there was less thickening inland relative to at the sample sites and the ice sheet margin.***

We agree with the above comment. As per the previous comment, we have added to the methodology for this section.

***Page 9, line 24: Unnecessary to abbreviate to MWP1A as only referred to one other time.***

Reference to MWP1A has been removed from the text.

***Page 9, line 35: Possibly "up to 655 m: : :".***

Agreed, we have changed this both here and in the Introduction, see response to earlier comment.

***In several places: Both ka and kyr is used. Pick one.***

We use ka when discussing a date, and kyr for a duration of time, following Aubry et al. (2009), Terminology of geological time: Establishment of a community standard. Stratigraphy, vol. 6, no. 2.

***References: Fogwill et al. (2014), and maybe other papers, are missing.***

We have added a reference to this paper and checked all other references.

***Fig. 1: Should 2a, 2b, : : : not be labelled 4a, 4b, : : :?***

Absolutely, we have updated the figure to reflect this.

***Fig. 3: What is the error envelope based on? What uncertainty has been used for the hypothetical sample concentrations? Typical analytical uncertainty?***

Typical analytical uncertainty used. We have added this to the figure caption (P14, lines 45 to 46).

***Supplementary material:***
***Would be useful if the study site figures had contours (e.g. from REMA).***

Contour lines (either 100 m or 200 m intervals) have been added to the supplementary figures. The colour of the contour lines varies due to the colour of the underlying satellite imagery.

***Would be nice to see a figure with the 14C ages plotted on a linear axis, perhaps vs the relative elevation of the samples.***

We have added supplementary figures for the 14C ages, plotted with their elevation above modern ice. These are also now referenced in the text.

**New Last Glacial Maximum Ice Thickness constraints for the Weddell Sea sector, Antarctica**

Keir A. Nichols[1], Brent M. Goehring[1], Greg Balco[2], Joanne S. Johnson[3], Andrew S. Hein[4], Claire Todd[5]

[1]. Department of Earth and Environmental Sciences, Tulane University, New Orleans, 70118, LA, USA.
[2]. Berkeley Geochronology Center, 2455 Ridge Road, Berkeley, 94709, CA, USA.
[3]. British Antarctic Survey, Natural Environment Research Council, High Cross, Madingley Road, Cambridge, CB3 0ET, UK.
[4]. School of GeoSciences, University of Edinburgh, Drummund Street, Edinburgh, EH8 9XP, UK.
[5]. Department of Geosciences, Pacific Lutheran University, Tacoma, 98447, WA, USA.

*Correspondence to:* Keir Nichols (knichol3@tulane.edu)

**Abstract.** This paper describes new Last Glacial Maximum (LGM) ice thickness constraints for three locations spanning the Weddell Sea Embayment (WSE) of Antarctica. Samples collected from the Shackleton Range, Pensacola Mountains, and the Lassiter Coast constrain the LGM thickness of the Slessor Glacier, Foundation Ice Stream, and grounded ice proximal to the modern Ronne Ice Shelf edge on the Antarctic Peninsula, respectively. Previous attempts to reconstruct LGM-to-present ice thickness changes around the WSE used measurements of long-lived cosmogenic nuclides, primarily $^{10}$Be. An absence of post-LGM apparent exposure ages at many sites led to LGM thickness reconstructions that were spatially highly variable, and inconsistent with flowline modelling. Estimates for the contribution of the ice sheet occupying the WSE at the LGM to global sea level since deglaciation vary by an order of magnitude, from 1.4 to 14.1 m of sea level equivalent. Here we use a short-lived cosmogenic nuclide, in situ produced $^{14}$C, which is less susceptible to inheritance problems than $^{10}$Be and other long-lived nuclides. We use in situ $^{14}$C to evaluate the possibility that sites with no post-LGM exposure ages are biased by cosmogenic nuclide inheritance due to surface preservation by cold-based ice and nondeposition of LGM-aged drift. Our measurements show that the Slessor Glacier was between 310 and up to 655 m thicker than present at the LGM. The Foundation Ice Stream was at least 800 m thicker, and ice on the Lassiter Coast was at least 385 m thicker than present at the LGM. With evidence for LGM thickening at all of our study sites, our in situ $^{14}$C measurements indicate that the long-lived nuclide measurements of previous studies were influenced by cosmogenic nuclide inheritance. Our LGM thickness constraints point toward a modest contribution from the WSE to global sea-level since deglaciation, with an estimated minimum contribution of <4.6 m, and possibly <1.5 m, based on the evidence for the lower limit of LGM thickness change at our study sites.

**1. Introduction**

This paper describes new in situ produced $^{14}$C derived Last Glacial Maximum (LGM) ice thickness constraints from three locations within the Weddell Sea Embayment (WSE) of Antarctica (Fig. 1). We broadly define the LGM as the period between ~15 and 25 ka when the Antarctic ice sheet volume was near its maximum extent in the last glacial-interglacial cycle. 
[revised manuscript text omitted]

[Figure]

Fig. 1

[Figure]

[Figure]

Fig. 2

[Figure]

[Figure]

Fig. 3

[Figure]

[Figure]

[Figure]

[Figure]

[Figure]

[Figure]

Fig. 4

[Figure]

[Figure]

Fig. 5

[Figure]

[Figure]

[Figure]

Fig. 6

[Figure]

[Figure]

Fig. 7

[Figure]

[Figure]

Fig. 8

[Figure]

5  Fig. 9

[Figure]

[Figure]

Fig. 10

[Figure]

Fig. 11

[Figure]

Fig. 12

**Supplement**

Table S1: Sample information. Photographs for samples (where available) available in ICE-D at http://antarctica.ice-d.org.

Table S2: In situ $^{14}$C analytical data and exposure ages. TUCNL column contains the Tulane University Cosmogenic Nuclide Laboratory code for each sample.

Figure S1: Sample locations at Mount Skidmore. Landsat 8 imagery courtesy of the U.S. Geological Survey. Contours at 100 m interval generated using the Reference Elevation Model of Antarctica (REMA; Howat et al., 2019).

Figure S2: Sample locations at Mount Provender. Landsat 8 imagery courtesy of the U.S. Geological Survey. Contours at 100 m interval generated using the Reference Elevation Model of Antarctica (REMA; Howat et al., 2019).

Figure S3: Sample locations on Mount Lampert and an unnamed nunatak, Lassiter Coast. Landsat 8 imagery courtesy of the U.S. Geological Survey. Contours at 100 m interval generated using the Reference Elevation Model of Antarctica (REMA; Howat et al., 2019).

Figure S4: Sample locations in the Schmidt Hills, Pensacola Mountains. Landsat 8 imagery courtesy of the U.S. Geological Survey. Contours at 200 m interval generated using the Reference Elevation Model of Antarctica (REMA; Howat et al., 2019).

Figure S5: Sample locations in the Thomas Hills, Pensacola Mountains. Two samples were collected within close proximity such that their markers overlap. Landsat 8 imagery courtesy of the U.S. Geological Survey. Contours at 200 m interval generated using the Reference Elevation Model of Antarctica (REMA; Howat et al., 2019).

Figure S6: Elevation versus in situ $^{14}$C age of samples from the Shackleton Range. Samples with in situ $^{14}$C concentrations equivalent to saturation are not shown.

Figure S7: Elevation versus in situ $^{14}$C age of samples from the Lassiter Coast.

Figure S8: Elevation versus in situ $^{14}$C age of samples from the Schmidt Hills. Only measurements yielding exposure ages ≤20 ka are presented.

Figure S9: Elevation versus in situ $^{14}$C age of samples from the Thomas Hills.

[Figure]

Fig. S1

[Figure]

[Figure]

Fig. S2

[Figure]

Fig. S3

[Figure]

[Figure]

[Figure]

Fig. S4

[Figure]

Fig. S5

[Figure]

[Figure]

[Figure]

Fig. S7

[Figure]

[Figure]

Fig. S9

---

## Author Comment (AC2) · 2 Aug 2019

We would like to thank Jane Andersen for the very thorough, thoughtful, and constructive review.

Text in bold and italics are comments made by the reviewer, with responses in regular text following.

Reviewer 2 Main comments

*My main concerns are as follows:*

*The replicate measurements demonstrate that there are serious problems with the reproducibility of this dataset. In the current form, this comes as a complete surprise halfway through the manuscript, as it is not reported in the abstract, introduction or conclusion were the results are stated without mentioning this important issue. I think this issue should be made more transparent throughout the manuscript…*

To address this, we now introduce the problem of the replicate measurements at the end of the Introduction and also discuss the rationale behind the replicates at the end of the Methods section, both prior to their first mention in the initial submission. We also now mention the issue in the Conclusion.

*…and that the implications for the non-replicated data should be discussed in more detail.*

We use the long-term replicate uncertainty (6%) to avoid the need to measure every sample in replicate.

*Given the authors' argument that the saturated samples are more prone to being erroneous due to contamination issues with modern carbon,*

Whilst spuriously high in situ $^{14}$C measurements have been reported, it is unlikely that samples would plot directly on the saturation concentration (or within the saturation concentration error envelope) when contaminated with modern carbon. See Fig. 1 here: https://www.geochronology-discuss.net/gchron-2019-7/. Bear in mind that the source of modern carbon for the linked study is from part of the quartz isolation process that was not used for the saturated sample in question. We don't think the samples exhibiting high scatter in their replicates were contaminated with modern carbon, but that the high scatter is sourced from elsewhere (grainsize, lithology, or another unidentified source).

*I am specifically concerned with whether too much emphasis is put on the single saturated sample from Mount Provender. Is there any evidence that this sample should be more reliably saturated than the replicated ones? This is particularly important since this is the only data point within the dataset that could otherwise provide a maximum ice sheet thickness during LGM.*

When we have observed anomalously high $^{14}$C concentrations from samples that were contaminated with modern carbon (see above response), in each case the resulting $^{14}$C measurement was far in excess of saturation, being completely overwhelmed with modern carbon. We haven't observed modern contamination from other sources (than the aforementioned study). In terms of why we think the one saturated sample at Mt. Provender is robust, we added the following in the Discussion (P8, lines 36 to 41):

"Whilst the upper limit of LGM ice at Mt. Provender is based on a single sample, we believe this sample is a reliable indicator of LGM ice thickness for the following reasons. The sample is sourced from

bedrock and therefore cannot have been subjected to geomorphic scenarios causing the exposure age to misrepresent the timing of ice retreat. Furthermore, froth flotation, which introduces modern carbon to sample material (Nichols and Goehring, in review), was not used to isolate quartz for this sample."

Both with this sample and for the study as a whole, we have tried to separate the evidence from the conclusions that we draw from said evidence, so that the reader can look at the evidence in isolation and come to their own conclusions.

*Furthermore, since this issue challenges the premise of the paper - that the maximum LGM thickness can be detected by the limit between saturated and unsaturated samples in an in situ 14C-elevation profile – I would like to see a brief discussion on how this issue could be handled in the future. Since measurements have been done on samples collected by others with the aim of dating with long-lived nuclides, such a discussion could further include a description of a sampling strategy that could better test your basic premise, would you e.g. recommend sampling bedrock for future applications of this method?*

Although we think a discussion on this topic is beyond the scope of the current paper, we do agree that a discussion on this would be useful in our response. Bedrock does remove potentially complicating geomorphic factors that may come with erratic samples. However, in our opinion the best sampling strategy would be erratic-bedrock pairs, but this is, of course, lithology and presence of erratics permitting. A boulder-erratic pair potentially permits the estimation for the onset of ice cover and the duration of the LGM – see Johnson et al. (2017) "The last glaciation of Bear Peninsula, central Amundsen Sea Embayment of Antarctica: Constraints on timing and duration revealed by in situ cosmogenic [14]C and [10]Be dating".

We wanted to make sure that we were as clear as possible in highlighting the problem of excess scatter in our data at the Schmidt Hills, and are glad that, from the review comments, it will now be even clearer. At this stage it is a subject of ongoing research by multiple research groups. Although not relevant to the samples in question in this present study, the following study looks into contamination from modern carbon (ps://www.geochronology-discuss.net/gchron-2019-7/) (the source of modern carbon in the paper was not used with the samples at the Schmidt Hills, however). Another avenue of research is the incomplete separation of quartz from surfaces in zeolites or clays. The samples in that study contaminated with modern carbon yielded in situ [14]C concentrations far in excess of saturation. There is no process that is expected to yield exactly-at-saturation concentrations except at random. Whilst it is evident that in the present state of [14]C chemistry some outliers are to be expected, it is extremely unlikely that an indistinguishable-from-saturation measurement can be explained by this process.

*2. I find the presentation and discussion of the implication of these data for the Weddell sea sector's contributions to LGM-present sea level somewhat misleading, and think the authors should give this subject some more consideration. Firstly, I think the estimated range of 2.2-5.8 m is somewhat deceiving as it gives the impression that this is a minimum-maximum envelope, while it is really based on minima estimates from two different locations. It should be clearly stated that these are both minimum estimates. Perhaps it would be more appropriate to make a single minimum estimate based on e.g. a smoothed surface fit to the minima-elevations.*

We agree that the original estimated range was inadvertently deceiving. The following is copied from our response to a comment from reviewer 1:

Agreed, we have altered the methodology for Sect. 4.4 (P10, lines 5 to 35). We have added "minimum" whenever describing the LGM thickness constraints and resulting SLE estimates. Furthermore, we have added to our original approach to produce a sea-level contribution estimate that is more quantitative than those presented in the initial submission. We now compare the modelled LGM ice thickness at each site from a range of ice sheet model outputs to our in situ $^{14}$C-derived ice thickness interpretations. We look at the sea-level equivalent for the Weddell sector for each ice sheet model output used. We have used the ice sheet model outputs of Le Brocq et al. (2011), Whitehouse et al. (2012) and Golledge et al. (2014). The LGM thickness constraints at all of our study sites are smaller than the predicted LGM thickness change of Golledge et al. (2014), which predicts a sea-level equivalent of ca. 4.6 m for the Weddell sector, placing an upper limit on our minimum SLE equivalent estimate. We also make it clear in the text that we provide an estimate for the minimum SLE for the WSE, and we now only use the minimum LGM ice thickness estimates for the sea-level contribution discussion. We have also added a new figure (12) to this section.

*Secondly, I don't see how the author's interpretation of a limited MWP-1A contribution from this sector of the AIS is based on the available cosmogenic data. In my opinion, the results provide robust minimum estimates of the maximum thickness during LGM, and Holocene thinning rates at sites where (sampled) nunatak elevations coincide with ice sheet surface lowering. Yet, given that the only maximum ice thickness estimate is based on a single saturated sample (which again is less certain owing to the replication issue mentioned above), nothing can be reliably said about thinning rates prior to the early Holocene.*

This issue was also brought up by reviewer 1 and on reflection we agree with both reviewers. We agree that the results provide minimum estimates at all sites, and possibly a maximum estimate for the Shackleton Range (see responses to reviewer 1 on this subject). We have removed any mention of MWP1A and any interpretation on when the majority of post-LGM thinning occurred.

*Specific comments:*

*P1, L20-22: specify already here that you use a 'short-lived' cosmogenic nuclide, which is less susceptible to inheritance problems than 10Be and other long-live nuclides.*

Added the suggested information to P1, lines 22 to 23.

*P2, L21: 'without erosion beneath cold-based ice' is confusing, perhaps 'when protected from erosion beneath cold-based ice'.*

Agreed, we have added the suggested text to P2, line 23.

*P2, L22: 'reduce concentrations' - concentrations will start to reduce immediately by decay when buried, specify 'to below measurable levels' or similar.*

Done, added to P2, line 26.

***P2, L24-25: briefly specify somewhere around this transition in the paper that in-situ 14C is less sensitive to inheritance due to its short half-live, this is key to understand the difference between using one nuclide or another. Also state half-lives of 10Be (+26Al) here or elsewhere.***

We have added "in situ produced 14C, a cosmogenic nuclide that is, owing to a short half-life of 5730 yr, largely insensitive to inheritance." to P2, line 30. We have also added the half-lives of 10Be and 26Al to P1, line 38.

***P2, L25&27: you are not really discussing and constraining 'LGM thickening', but the maximum thickness and subsequent thinning.***

We agree that "LGM thickening" could be confusing. We have changed this phrase to say that we are constraining the LGM thickness throughout the document.

***P3, L19: again, 'reduce' on its own is confusing, add 'to below analytical limit' or similar. Also specify what you mean by 'short' by stating the approximate burial time it would typically require.***

We addressed this same topic which was brought up by reviewer 1 regarding the same sentence. We expand on the response below:

In this instance, what we meant was that the in situ $^{14}$C concentration is significantly reduced, but when talking about the time to reach beneath analytical limits this would depend on the production rate at the site and initial concentration. We have added the word "significantly" before "reduce", and added an example:

"For example, a burial duration beneath non-erosive, cold-based ice of 11 kyr results in ca. 74% of the original in situ 14C decaying away." (P3, line 26 to 27).

***P3, L21: 'surfaces not covered by ice during the LGM', this could be stated more generally, e.g. 'continuously-exposed, slow-eroding surfaces'***

Text altered accordingly, see P3, line 28.

***P3, L22: it would also require low erosion rates to reach such high apparent ages, specify this.***

Added to P3, line 29.

***P3, L25: cold-based ice has been shown to sometimes erode (e.g. Atkins et al., 2002, Geology v30 p.659-662). Perhaps specify 'cold-based, non-erosive ice' or simply 'nonerosive ice'***

Added "non-erosive" to P3, line 32.

***P3, L38 – P4, L4: I think the discussion of whether the erratics record the history of their sampling location could be made more concise. It shouldn't matter for in-situ 14C whether the erratic has been***

*'covered and uncovered' in place or during transport, as long as it has been buried long enough and then deposited by the ice, with no subsequent movement? Also avoid moving back-and-forth between discussing bedrock and erratics.*

We want to include the text in this section because we wanted to briefly discuss potential limitations of the method, as well as introduce some concepts that we discuss later on in the paper when we assess the in situ 14C data (Sect. 4.1). In particular we wanted to introduce the observation of Balco et al. (2019) of mass movement and supraglacial transport potentially resulting in saturated in situ 14C concentrations collected from lower elevations than finite ages, which goes against the assumption that erratics and bedrock will always yield the same information with respect to ice thinning when using in situ 14C.

*P3, L46: specify which assumption.*

We added "that bedrock and erratic samples provide the same information with respect to the timing of changes in ice thickness" to P4, line 6 to 7.

*P4, L9-11: I don't think it is necessary to repeat this information here*

We removed the text.

*P6, L. 43-46: At present this section has a bit of an 'ad-hoc argument' feel to it, I think it would be better to first provide an objective discussion on what criteria can be used to assess the validity of this method before you go through each site.*

We have added a few sentences to the start of this section to introduce the section. The following text is now on P7, lines 15 to 19:

"To assess the validity of this method, we can, for example, identify where the in situ 14C data records ice thinning, with saturated samples or the oldest exposure ages at the highest elevations and a trend of decreasing in situ 14C age toward modern ice surfaces. Consistency between in situ 14C data and other nuclide concentrations (e.g. 10Be) could also help validate the in situ 14C measurements. We also look at factors beyond the in situ 14C concentrations, such as the glaciological link between study sites, which may add clarity where the in situ 14C measurements show a high degree of scatter."

*P6, L. 44: I don't think the Lassiter coast dataset can be strictly described as having a 'linear age-elevation relation', as it requires at least two lines two fit the data. Perhaps rephrase to 'ages continuously decrease towards the present ice sheet surface'.*

We have altered the text accordingly.

*P7, L7: 008-NNS doesn't seem to be replicated here? I don't find a saturated (replica-) measurement in the table or figure.*

This sample, 008-NNS, was measured once in this study, and also once in the study by Balco et al. (2016). We altered the text (P7, lines 31 to 33) to hopefully make this clearer.

***P7, L12-13: You don't present replicate measurements from elsewhere than the Schmidt hills, so this sentence reads a bit strange.***

To address this, we have changed the sentence to:
"Why the replicate measurements from samples from the Schmidt Hills display a high degree of scatter remains to be determined." (P7, lines 36 to 37).

***P7, L14: This transition was not obvious to me. This also goes for the next two paragraphs, I think you could link the various paragraphs discussing the poor replicability better. The 'Regardless,' makes it sounds like you are moving on to another subject. This paragraph is discussing whether there could be any geological reasons that the in-situ 14C profiles are 'inverted'?***

To add clarity to this section we have altered the transitions between the paragraphs to help the section flow better and have also changed the order of the paragraphs to an order that is more logical. We also expanded on the sentence that started with "Regardless" so that it now reads:

"Regardless of the cause of the high degree of scatter observed in the replicate measurements, we need to discuss possible explanations for apparently infinite ages at lower elevations than apparently finite ages to isolate which measurements (infinite vs finite replicates) are the most valid to base interpretations on." (P8, lines 3 to 5).

***P7, L22: specify 'erratic' samples***

Added "erratic".

***P7, L26-27: I couldn't follow this argument.***

We agree that this was confusing, and have altered the text (P8, lines 16 to 18):

"Whilst this could explain the low-elevation saturated sample at Mt. Skidmore, as well as infinite measurements situated beneath finite measurements at the Schmidt Hills, it does not explain the poor reproducibility of the Schmidt Hills measurements."

***P7, L37: I think you need to discuss here whether or not you expect the same issue for the saturated, but non-replicated sample from Mount Provender. Given only two data points from this site, the fact that the 'profile' isn't inverted is not a very strong argument, or?***

We agree that the fact that the profile of two data points at Mt. Provender is not a strong argument, which we think is adequately covered by the preceding phrase "Though limited by the number of samples". Regarding the saturated sample at Mt. Provender, we now explain our thoughts on the robustness of this measurement in the Discussion (see our response to the first main concern of reviewer 1). In the introduction and conclusion, we use the phrase "up to 655 m" in the Introduction and Conclusion (P1, line 24, and P11, line 1). We also highlight the limitation in the Results (P6, lines 21 to 25) and the Discussion. Despite the reasons to believe it is a robust measurement, we only use the minimum LGM ice thickness constraints for the sea-level contribution discussion.

*P8, L11: provide the in situ 14C ages with uncertainty here to make it easy for the reader to compare*

We added the ages to the text.

*P9, L23: You first state that your data 'do not preclude a significant contribution earlier than (the early to mid-Holocene)', yet move directly on to suggest that MWP1A was not significant in this region, thus contra-arguing yourself. I don't think you can constrain MWP1A thinning on the basis of these data.*

We agree and have removed any reference to MWP1a, see above responses.

*Fig. 1: boxes should refer to Fig 4a-d, not 2a-d.*

*We have altered the labels of the boxes accordingly.*

*Fig. 2: legend P13, L13: remove 'constraints' before 'ice'*

Altered accordingly.

*Fig. 3: specify that you used zero erosion (presumably) for these calculations. It looks strange to me that the saturated samples are labelled as 'true' exposure. The figure is a bit blurry compared to the others.*

We have added "assuming no surface erosion" to the caption. By "true" we mean a concentration that is equivalent to the ice history in the left plot. We agree that using "age" is strange when it includes saturated samples, so we changed the caption to reflect this. Now reads:

" "True exposure" refers to the resulting 14C concentration associated with the ice surface change history on the left plot."

We updated the Figure so that it is the full resolution, thank you for highlighting this.

*Fig. 4: green color of sample points in legend and on map does not appear to be the same*

We have altered the figure so that the green colour is the same for the sample points in the map and the legend.

*Fig. 5: add legend showing circle=erratic, triangle=bedrock.*

We will make sure this is added in the final version.

*Fig. 9+10: legend refers to maps on left, but they are on the right*

Text changed accordingly.

*P1, L32: Should Antarctic ice sheet(s) be in plural, west+east?*

We think it would be fair to use the plural, but we stick to the singular when referring to both the east and west ice sheets at the same time. See Bentley and Anderson (1998) whom refer to the ice sheets individually as well as collectively as the "Antarctic ice sheet" in the first two paragraphs of their introduction. Also see the title of Hillenbrand et al. (2014) "Reconstruction of changes in the Weddell Sea sector of the Antarctic Ice Sheet since the Last Glacial Maximum".

*P2, L18: specify 'non-erosive' burial*

We have added this to the text.

*P3, L22: 'measured today' seems unnecessary*

Agreed, we removed this from the text.

*P4, L31: remove 'from'*

Altered text accordingly.

*P5, L13: 'analysed' and 'analysis' in same short sentence*

We changed "analysed" to "used".

*P6, L18: perhaps 'ice thinning' rather than 'deglaciation'?*

Agreed, we changed it to ice thinning.

*P7, L17: fig 10 referred before fig. 9. Fontsize is odd*

We have changed the text so that Fig. 10 is referred to after Fig. 9, and fixed the font size. (now Figs. 10 and 11 with the addition of a new figure).

*P10, L6-7: BG rather than BM?*

Absolutely, changed to fix this error.

*Supplementary table S2: Spreadsheet tab is named 'Table 3' while table is 'Table 2'.*

We changed the name of the tab in Table S2.

**New Last Glacial Maximum Ice Thickness constraints for the Weddell Sea sector, Antarctica**

Keir A. Nichols[1], Brent M. Goehring[1], Greg Balco[2], Joanne S. Johnson[3], Andrew S. Hein[4], Claire Todd[5]

[1]. Department of Earth and Environmental Sciences, Tulane University, New Orleans, 70118, LA, USA.
[2]. Berkeley Geochronology Center, 2455 Ridge Road, Berkeley, 94709, CA, USA.
[3]. British Antarctic Survey, Natural Environment Research Council, High Cross, Madingley Road, Cambridge, CB3 0ET, UK.
[4]. School of GeoSciences, University of Edinburgh, Drummund Street, Edinburgh, EH8 9XP, UK.
[5]. Department of Geosciences, Pacific Lutheran University, Tacoma, 98447, WA, USA.

*Correspondence to:* Keir Nichols (knichol3@tulane.edu)

**Abstract.** This paper describes new Last Glacial Maximum (LGM) ice thickness constraints for three locations spanning the Weddell Sea Embayment (WSE) of Antarctica. Samples collected from the Shackleton Range, Pensacola Mountains, and the Lassiter Coast constrain the LGM thickness of the Slessor Glacier, Foundation Ice Stream, and grounded ice proximal to the modern Ronne Ice Shelf edge on the Antarctic Peninsula, respectively. Previous attempts to reconstruct LGM-to-present ice thickness changes around the WSE used measurements of long-lived cosmogenic nuclides, primarily $^{10}$Be. An absence of post-LGM apparent exposure ages at many sites led to LGM thickness reconstructions that were spatially highly variable, and inconsistent with flowline modelling. Estimates for the contribution of the ice sheet occupying the WSE at the LGM to global sea level since deglaciation vary by an order of magnitude, from 1.4 to 14.1 m of sea level equivalent. Here we use a short-lived cosmogenic nuclide, in situ produced $^{14}$C, which is less susceptible to inheritance problems than $^{10}$Be and other long-lived nuclides. We use in situ $^{14}$C to evaluate the possibility that sites with no post-LGM exposure ages are biased by cosmogenic nuclide inheritance due to surface preservation by cold-based ice and nondeposition of LGM-aged drift. Our measurements show that the Slessor Glacier was between 310 and up to 655 m thicker than present at the LGM. The Foundation Ice Stream was at least 800 m thicker, and ice on the Lassiter Coast was at least 385 m thicker than present at the LGM. With evidence for LGM thickening at all of our study sites, our in situ $^{14}$C measurements indicate that the long-lived nuclide measurements of previous studies were influenced by cosmogenic nuclide inheritance. Our LGM thickness constraints point toward a modest contribution from the WSE to global sea-level since deglaciation, with an estimated minimum contribution of <4.6 m, and possibly <1.5 m, based on the evidence for the lower limit of LGM thickness change at our study sites.

**1. Introduction**

This paper describes new in situ produced $^{14}$C derived Last Glacial Maximum (LGM) ice thickness constraints from three locations within the Weddell Sea Embayment (WSE) of Antarctica (Fig. 1). We broadly define the LGM as the period between ~15 and 25 ka when the Antarctic ice sheet volume was near its maximum extent in the last glacial-interglacial cycle. 
[revised manuscript text omitted]

[Figure]

Fig. 1

[Figure]

[Figure]

Fig. 2

[Figure]

[Figure]

Fig. 3

[Figure]

[Figure]

[Figure]

[Figure]

[Figure]

[Figure]

Fig. 4

[Figure]

[Figure]

Fig. 5

[Figure]

[Figure]

[Figure]

Fig. 6

[Figure]

Fig. 7

[Figure]

[Figure]

Fig. 8

[Figure]

[Figure]

5   Fig. 9

[Figure]

[Figure]

Fig. 10

[Figure]

Fig. 11

[Figure]

Fig. 12

**Supplement**

Table S1: Sample information. Photographs for samples (where available) available in ICE-D at http://antarctica.ice-d.org.

Table S2: In situ $^{14}$C analytical data and exposure ages. TUCNL column contains the Tulane University Cosmogenic Nuclide Laboratory code for each sample.

Figure S1: Sample locations at Mount Skidmore. Landsat 8 imagery courtesy of the U.S. Geological Survey. Contours at 100 m interval generated using the Reference Elevation Model of Antarctica (REMA; Howat et al., 2019).

Figure S2: Sample locations at Mount Provender. Landsat 8 imagery courtesy of the U.S. Geological Survey. Contours at 100 m interval generated using the Reference Elevation Model of Antarctica (REMA; Howat et al., 2019).

Figure S3: Sample locations on Mount Lampert and an unnamed nunatak, Lassiter Coast. Landsat 8 imagery courtesy of the U.S. Geological Survey. Contours at 100 m interval generated using the Reference Elevation Model of Antarctica (REMA; Howat et al., 2019).

Figure S4: Sample locations in the Schmidt Hills, Pensacola Mountains. Landsat 8 imagery courtesy of the U.S. Geological Survey. Contours at 200 m interval generated using the Reference Elevation Model of Antarctica (REMA; Howat et al., 2019).

Figure S5: Sample locations in the Thomas Hills, Pensacola Mountains. Two samples were collected within close proximity such that their markers overlap. Landsat 8 imagery courtesy of the U.S. Geological Survey. Contours at 200 m interval generated using the Reference Elevation Model of Antarctica (REMA; Howat et al., 2019).

Figure S6: Elevation versus in situ $^{14}$C age of samples from the Shackleton Range. Samples with in situ $^{14}$C concentrations equivalent to saturation are not shown.

Figure S7: Elevation versus in situ $^{14}$C age of samples from the Lassiter Coast.

Figure S8: Elevation versus in situ $^{14}$C age of samples from the Schmidt Hills. Only measurements yielding exposure ages ≤20 ka are presented.

Figure S9: Elevation versus in situ $^{14}$C age of samples from the Thomas Hills.

[Figure]

Fig. S1

[Figure]

[Figure]

Fig. S2

[Figure]

Fig. S3

[Figure]

[Figure]

Fig. S4

[Figure]

[Figure]

Fig. S5

[Figure]

[Figure]

Fig. S6

[Figure]

Fig. S7

[Figure]

[Figure]

Fig. S9

---

## Author Response (AR1)

Thank you for the suggested revisions. We have incorporated all revisions suggested by the Editor, as can be seen in the revised manuscript below. Please note the change to the title of the manuscript (as suggested), as well as the corrected middle initial for one of the authors (Andy S. Hein).

List of changes made in the manuscript:

All references to "sea-level" have been changed to "sea level".
Removed all references to the "offshore" shelf edge, referring to it as simply the "shelf edge".
All "m asl" changed, to "m a.s.l." and are now consistent.
We added references for the $^{10}$Be and $^{26}$Al half-lives (page 1, lines 35-36).
Replaced "suggests" with "indicates" in all suggested instances.
Replaced "greater" with "larger" in all suggested instances.
Page 4, line 44 and page 5, lines 1 and 2 now make reference to Fig. 4b.
Page 5, line 10 now makes reference to Fig. 1.
All problems with the reference list that were highlighted by the Editor have been rectified.
Figs. 1, 2, 3, and 9 and 12 have been corrected as per the suggestions from the Editor.

Specific changes to the text are highlighted with blue.

Page 1, line 2:

"sector" changed to "embayment".

The following three parts of the manuscript have been altered to make the information clearer:

Page 1, lines 27 to 29:

[revised manuscript text omitted]

**Additional changes**

In addition, we have added grey shading to Fig. 3 and Figs. 5 to 8. Thanks to the review comments on another paper that we have submitted, a mistake in our uncertainties for the values in the column "total $^{14}$C atoms blank corrected" was brought to our attention. We have updated the values in the column in Table S2 (column "S" when opened in Microsoft Excel). Thankfully, the changes in the values have no influence on the final $^{14}$C concentrations or their associated uncertainties. In Table S2 we have also altered some headings that previously incorrectly used the term "error", replacing it with "±1σ" where appropriate. We also added units to the cell "O5", which was omitted in our initial submission. We also added the word "from" to the acknowledgements, page 11, line 20.

[revised manuscript text omitted]

Fig. 1

[Figure]

[Figure]

Fig. 2

[Figure]

[Figure]

Fig. 3

[Figure]

[Figure]

[Figure]

[Figure]

[Figure]

Fig. 4

[Figure]

Fig. 5

[Figure]

[Figure]

Fig. 6

[Figure]

[Figure]

[Figure]

Fig. 7

[Figure]

[Figure]

Fig. 8

[Figure]

Fig. 9

[Figure]

[Figure]

Fig. 10

[Figure]

Fig. 11

[Figure]

Fig. 12

[Figure]

Moved (insertion) [1]

Moved up [1]: